# Modeling Fine-Grained Hand-Object Dynamics for Egocentric Video Representation Learning

**Baoqi Pei**[1,2*], **Yifei Huang**[3,2*], **Jilan Xu**[4,2], **Guo Chen**[5], **Yuping He**[5], **Lijin Yang**[3],
**Yali Wang**[6,2], **Weidi Xie**[7,2], **Yu Qiao**[2], **Fei Wu**[1], **Limin Wang**[5,2]
[1]Zhejiang University, [2]Shanghai Artificial Intelligence Laboratory, [3]The University of Tokyo,
[4]Fudan University, [5]Nanjing University, [6]SIAT, [7]Shanghai Jiao Tong University,
`peibaoqi@gmail.com; hyf@iis.u-tokyo.ac.jp`

## Abstract

In egocentric video understanding, the motion of hands and objects as well as their interactions play a significant role by nature. However, existing egocentric video representation learning methods mainly focus on aligning video representation with high-level narrations, overlooking the intricate dynamics between hands and objects. In this work, we aim to integrate the modeling of fine-grained hand-object dynamics into the video representation learning process. Since no suitable data is available, we introduce **HOD**, a novel pipeline employing a hand-object detector and a large language model to generate high-quality narrations with detailed descriptions of hand-object dynamics. To learn these fine-grained dynamics, we propose **EgoVideo**, a model with a new lightweight motion adapter to capture fine-grained hand-object motion information. Through our co-training strategy, EgoVideo effectively and efficiently leverages the fine-grained hand-object dynamics in the HOD data. Extensive experiments demonstrate that our method achieves state-of-the-art performance across multiple egocentric downstream tasks, including improvements of **6.3%** in EK-100 multi-instance retrieval, **5.7%** in EK-100 classification, and **16.3%** in EGTEA classification in zero-shot settings. Furthermore, our model exhibits robust generalization capabilities in hand-object interaction and robot manipulation tasks. Code and data are available at `https://github.com/OpenRobotLab/EgoHOD/`.

## 1 Introduction

Egocentric video understanding has recently garnered increasing attention due to its crucial role in areas such as augmented reality (Pan et al., 2023), embodied AI (Srivastava et al., 2022; Huang et al., 2024b), and personalized assistants (Huang et al., 2018). With the collection of large-scale egocentric video datasets (Damen et al., 2020; Grauman et al., 2022), researchers begin to adopt video-language pretraining (Lin et al., 2022) based on these annotations to learn egocentric video representations. Since the original annotations tend to be highly template-driven and lack diversity, previous works explore using Large Language Models (LLM) to rephrase the narration (Zhao et al., 2023) or introducing new video-language pairs from exocentric datasets (Dou et al., 2024). This scheme has shown its success in a wide range of downstream tasks (Plizzari et al., 2024).

However, as can be seen from the example in Figure 1 right, the original annotations in egocentric video datasets are typically highly condensed, describing only overall actions like "C draws on a book" or "C moves both hands". Since no additional information is provided, previous works like LaViLa (Zhao et al., 2023) can only rephrase at the same level of abstraction as the original annotations, neglecting a crucial aspect of egocentric videos – the fine-grained dynamics of hands and objects. Most egocentric videos contain a large portion of hand-object interactions, which reflects the camera wearer's behavior and intentions. As will be seen, integrating this information in vision-

---

*Equal contribution. Yifei Huang is the corresponding author.

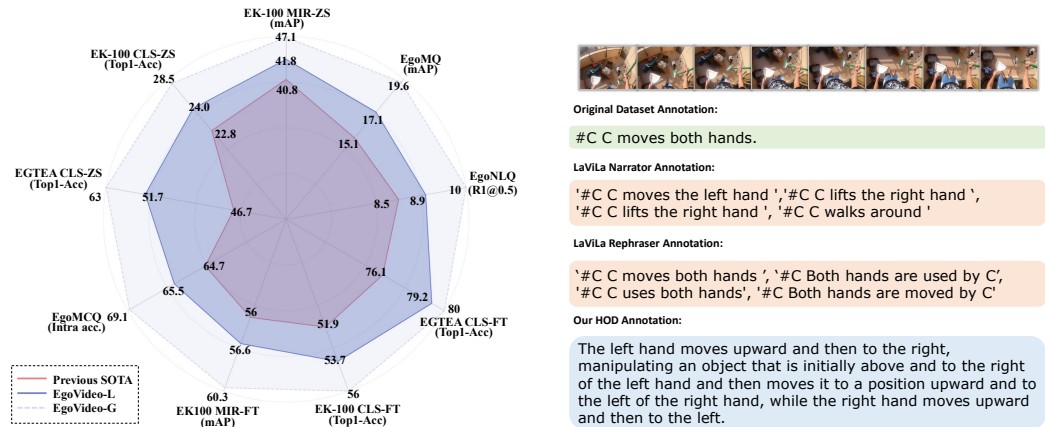

Figure 1: *Left*: Our **EgoVideo** model achieves state-of-the-art performance across multiple video benchmarks by learning fine-grained hand-object dynamics from videos. *Right*: Annotations from different sources: original Ego4D annotation (Grauman et al., 2022), LaViLa (Zhao et al., 2023), and our **HOD**. Our HOD annotations provide a detailed description of hand movements and object manipulation, demonstrating a higher level of detail and context.

language pretraining significantly enhances egocentric video representation learning, resulting in state-of-the-art performance across various benchmarks.

Firstly, to incorporate hand-object dynamics into vision-language pretaining, it is essential to construct data that accurately captures the detailed motion of hands and objects in videos. A recent work directly uses the output of off-the-shelf hand-object detectors (Shan et al., 2020) as the ground truth of auxiliary targets in the pretraining (Zhang et al., 2023). However, this approach only models the appearance of hands and objects without considering their dynamics. It also fails to learn the semantic connections between hand-object interactions and the original narration. To address this, we introduce **HOD**, a novel framework for generating descriptions with fine-grained **H**and-**O**bject **D**ynamics for a given video clip. We begin with using hand-object detectors to obtain bounding boxes of hands and contact objects. Then we design prompts based on these bounding boxes to generate descriptions of the trajectories of the hand and object, as well as their contact states and positions. Finally, using the new prompts and original annotations, we leverage a large language model (LLM) to generate semantically rich captions that encompass the motion states of hands and objects. By utilizing high framerate inputs, we ensure the capture of more detailed motions.

Secondly, to efficiently and effectively exploit the fine-grained spatiotemporal information in HOD, we propose **EgoVideo**, a novel ViT-based model with a lightweight motion adapter. Cooperating with the HOD data, EgoVideo employs a dual-branch design and co-training strategy. The backbone branch is trained normally to learn fundamental video-language alignment, while the adapter branch is trained with a higher framerate to capture detailed hand-object dynamics. The motion adapter has a separable convolution design, allowing for information aggregation from both adjacent frames temporally and from hands and objects at different locations. This design enables EgoVideo to model detailed hand-object dynamics while maintaining low computational costs. This also allows us to scale the model size to 1B to fully unlock its potential to comprehend egocentric videos.

We extensively evaluate EgoVideo across multiple pretraining data sources and various egocentric downstream tasks. Experimental results show that our model sets a new state-of-the-art on 9 tasks as partially shown in Figure 1. Notably, our model also achieves the best performance under the same model size in both zero-shot and fine-tuning settings. Further experiments demonstrate that our HOD data is also beneficial for robot manipulation tasks.

Our main contributions are as follows: (1) We develop a HOD data pipeline to generate captions that describe fine-grained hand-object dynamics, which are crucial for egocentric video understanding; (2) We propose EgoVideo, a dual-branch model with a novel lightweight motion adapter and a co-training strategy to leverage the HOD data efficiently and effectively; (3) We demonstrate state-of-the-art performance on 12 downstream tasks, and our approach generalizes well to robot manipulation tasks. All code and data will be made publicly available.

## 2 RELATED WORK

**Egocentric Video Understanding** is receiving increasing research attention. Previous works focus on diverse tasks such as action recognition (Plizzari et al., 2022; Huang et al., 2020a), action anticipation (Girdhar & Grauman, 2021), and cross-view understanding (Xue et al., 2022; Huang et al., 2024a; Luo et al., 2024). Recent methods begin to work on egocentric representation learning (Lin et al., 2022; Pei et al., 2024) using the large-scale data from Ego4D (Grauman et al., 2022), or refining the Ego4D narrations by LLM rephrasing (Zhao & Krähenbühl, 2023). A recent work also searches for additional data from exocentric datasets to improve the pretraining (Dou et al., 2024). However, since the Ego4D narrations are highly abstract, these methods fail to learn one critical aspect of egocentric videos – fine-grained hand-object dynamics. Recently, Helping Hands (Zhang et al., 2023) utilizes hand and object coordinates as auxiliary targets during pretraining. However, it only focuses on the spatial information of hands and objects, neglecting their motion dynamics. Additionally, the provided supervision does not integrate the states of hands and objects with the video descriptions, limiting the model's ability to comprehend fine-grained details.

Unlike previous works, we propose the first method to integrate the hand-object dynamics into egocentric representation learning. On the data side, we propose the **HOD** (Hand-Object Dynamics) pipeline, which generates high-quality video-language pairs. The language in these pairs explicitly represents the complex states and motions of hands and objects in the videos, enabling the model to learn detailed information about these dynamics. On the model side, we introduce **EgoVideo**, a model equipped with a lightweight motion adapter. This adapter is designed to effectively capture the intricate hand and object dynamics provided by the HOD data, enhancing the model's ability to understand and interpret fine-grained dynamics in egocentric videos.

**Video-Language Representation Learning** has also attracted researchers after the success of CLIP (Radford et al., 2021), due to the need for generating robust video representations. Several large-scale video-language datasets (Kay et al., 2017; Miech et al., 2019; Caba Heilbron et al., 2015) further fueled the research in this area. However, generating high-quality video-text pairs remains a challenging task, prompting researchers to develop innovative solutions. LaViLa (Zhao et al., 2023) leverages Large Language Models (LLMs) to generate dense narrations for videos. Video Recap (Islam et al., 2024) utilizes a curriculum learning training scheme to generate summaries for long videos. EMBED (Dou et al., 2024) and EgoInstructor (Xu et al., 2024) use rules or retrieval models to add additional training data. However, the previous methods can only pretrain their models at the same abstraction level as the original annotation. In contrast, our approach integrates finer-level details into the representation learning process.

**Hand-Object Interaction Understanding** has long been a key research topic within the field of egocentric vision. In recent years, several works have made significant strides in modeling estimate 3D hand joints (Brahmbhatt et al., 2020; Cai et al., 2018; Yang & Yao, 2019; Yuan et al., 2018; Ohkawa et al., 2023) and reconstructing hand-object shape (Cao et al., 2021; Doosti et al., 2020; Hasson et al., 2019; 2020; Liu et al., 2021). EgoHOS (Zhang et al., 2022) provides a labeled dataset with fine-grained per-pixel labels of hand and objects and a reliable foundational tool for 2D hand-object segmentation, 100DOH (Shan et al., 2020) introduces a large-scale video dataset containing hands and hand-object interactions, providing a rich resource for hand object detector training. In our work, we utilize existing hand and object detectors in our HOD pipeline to convert information related to hand/object motion and contact details into natural language descriptions. By integrating these detailed descriptions with our EgoVideo model, we can integrate this finer level of detail into the video representation learning process.

## 3 METHOD

### 3.1 DATA GENERATION PIPELINE: HOD

The fine-grained dynamics of hands and objects play a pivotal role in egocentric video understanding (Fathi et al., 2011a). To effectively integrate this information in the video-language pretraining process, we propose HOD, a novel data generation pipeline to transform hand-object dynamics into natural languages. An overview of HOD is illustrated in Figure 2 top. First, we utilize an off-the-shelf hand object detector (Shan et al., 2020) to generate bounding boxes for hands and objects in

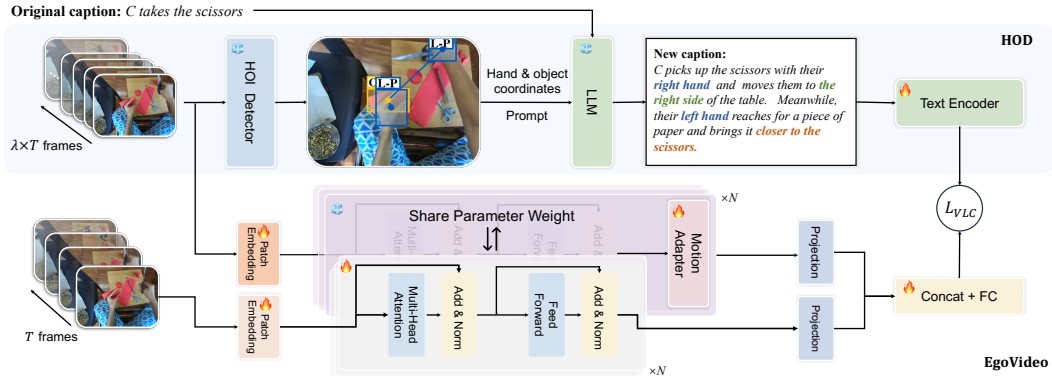

Figure 2: Illustration of our **HOD** pipeline and **EgoVideo** model. In our **H**and-**O**bject **D**ynamics data generation pipeline (top), we first use a hand object detector to obtain the spatial coordinates of hands and objects in the clip, then we combine the motion information of hands and objects with the original narrations to generate semantically richer narrations. In our **EgoVideo** model (bottom), the backbone is trained with a lower framerate. We design a lightweight motion adapter to learn fine-grained dynamics efficiently with higher framerate inputs.

each frame of the video clips. Next, we employ a large language model (AI et al., 2024) to enrich the original video captions. The model is prompted to generate new narrations that integrate the original captions with hand-object dynamics information, enhancing the semantic richness of the annotations. Below, we go into the details of the HOD data generation process.

### 3.1.1 DATA SELECTION

Before going into the generation process, it is essential to select appropriate source data. The basic data component comes from the 4M subset (Lin et al., 2022) of Ego4D (Grauman et al., 2022) which has been proven useful in egocentric video-language pretraining (Pramanick et al., 2023). Additionally, we curate data from the large-scale HowTo100M dataset (Miech et al., 2019) since it contains rich hand-object interactions. We specifically choose How2-Interlink7M (Wang et al., 2024a) which contains 7M clips with high-quality GPT-4 (OpenAI, 2023) refined caption. Since the videos are from diverse sources that may include a portion that impedes egocentric representation learning, we employ a filtering technique to retain only clips with egocentric style. We train a style classifier $\mathcal{P}$ by manually annotating 10,000 clips as "ego-like" or "non-ego-like". With this classifier, we obtain an additional 3.4M egocentric-style clips. More details can be found in Appendix A.

### 3.1.2 GENERATING CAPTIONS WITH HAND OBJECT DYNAMICS

In this section, we introduce our HOD framework. Existing methods of refining descriptions in video-language pretraining(Zhao et al., 2023; Dou et al., 2024) focus on high-level abstracts but overlook the fine-grained details of hand-object dynamics. This oversight is detrimental in egocentric representation learning, where understanding these interactions forms a considerable proportion of egocentric videos by nature. To address this gap, in our HOD framework, we first detect the positions of hands and objects using a hand object detector. With this information, we prompt a large language model to augment the original annotation with detailed descriptions of hand and object movements. Followed by the subsequent video-language pretraining, our EgoVideo model can understand videos at a finer-grained level.

**Hand Object Dynamics Detector.** Thanks to the rapid advancement in the field of hand-object interaction (Jiang et al., 2021; Ohkawa et al., 2023), off-the-shelf hand-object detectors can provide robust hand and object positions. In our framework, we employ 100DOH (Shan et al., 2020) as the detector $\Phi_{\text{det}}$ for bounding boxes extraction.

For a video clip $x = (x_1, x_2, ...x_T)$, we uniformly sample $n = 16$ frames within the clip to obtain fine-grained motion information. Then we use $\Phi_{\text{det}}$ to acquire the bounding boxes of hands and objects on these frames, which can be represented as

$$LH_i, RH_i, LO_i, RO_i = \Phi_{\text{det}}(x_i) \tag{1}$$

where $LH_i, RH_i, LO_i, RO_i$ denotes the bounding box of the left hand, right hand, objects in contact with the left hand, and objects in contact with the right hand in the $i$-th frame. We use linear interpolation to compensate for missing hand boxes of frame $t$ if the corresponding hand boxes can be detected for both frames $x_{t-1}$ and $x_{t+1}$.

**Hand Object Dynamics Rephraser.** Current pretraining methods only use high-level language descriptions (*e.g.,* "C takes the scissors" in Figure 2), which lacks important egocentric details like hand and object interaction. In this work, we incorporate these details into the video-language pretraining process. Hand-object dynamics encompass a variety of information, including bounding boxes of hands and objects, hand and object movement directions and trajectories, as well as their contact conditions. To integrate all this information into the video-language pretraining process, we use a LLM as a rephraser to express these dynamics in natural language.

Specifically, we employ Yi-34B (AI et al., 2024) as our LLM. To capture the nuances of hand and object movements, we extract the central points of bounding boxes to derive trajectories for hands and objects. This process yields six essential categories of information: spatial-temporal data for 1) the left hand, 2) the right hand, 3) objects contacted by the left hand, 4) objects contacted by the right hand, 5) objects contacted by both hands, and 6) the original narration. We then prompt the LLM to amalgamate this detailed information, enabling the generation of rich narratives that intricately describe hand-object dynamics. Further details on prompting can be found in Appendix A.

**Analysis of HOD Data.** We conduct additional analyses on our HOD data to evaluate its quality. First, we identify the top 30 most frequent words in HOD captions and the original EgoClip narrations and plot their normalized frequencies in Figure 3. The EgoClip narrations exhibit a more pronounced long-tail distribution, while our HOD captions display a more balanced distribution. Notably, HOD captions include many "dynamic" words, such as "up" and "downwards," which aligns with the rationale behind our data generation process. To further verify the quality of our HOD, we employ GPT-4o (OpenAI, 2023) for quality assessment. We randomly select 1000 clips and let GPT evaluate the score of the caption data for the video clip in a range from 0 to 10. To ensure GPT does not simply assign high scores based on the length of the captions, we also conduct random gerund replacements on our data for comparison. The results, summarized in Table 1, show that our HOD data have a significantly better GPT-Score. Additional details on the scoring process and evaluations using other metrics are provided in Appendix A.

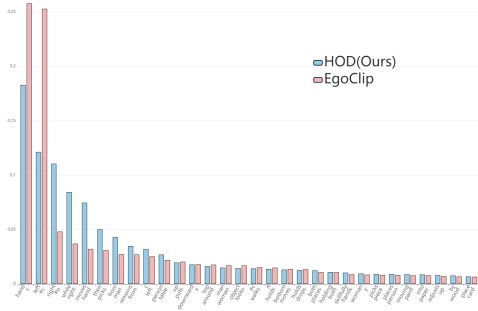

Figure 3: Normalized frequency of the Top-30 word in EgoClip (green) and our HOD (blue). Our HOD data has a less long-tail distribution, showing its word diversity.

Table 1: Results of narration quality, where HOD-random represents the narration after replacing keywords. The GPT-Score ranges from 0 to 10, with higher values indicating higher quality of narration.

| Data | GPT-Score |
|------|-----------|
| EgoClip | 5.53 |
| HOD-random | 3.70 |
| **HOD** | **7.71** |

## 3.2 EGOCENTRIC REPRESENTATION LEARNING MODEL: EGOVIDEO

The narrations generated by our HOD pipeline are highly detailed. As a result, the previous pretraining scheme struggles to capture the corresponding visual information at this level of detail. In response, we introduce EgoVideo (Figure 2 bottom), a model comprising a backbone and a motion adapter. The motion adapter aids in learning fine-grained hand-object dynamics from densely sampled video frames. Cooperating with a co-training strategy, our EgoVideo model can obtain richer video representations while maintaining computational efficiency.

**Visual and Text Encoder.** Following the standard video-language pretraining setting (Lin et al., 2022), our model includes a visual encoder $\mathcal{F}_v$ (including our motion adapter) and a text encoder

$\mathcal{F}_t$. In the visual encoder, for a clip $x \in \mathbb{R}^{T \times H \times W \times 3}$, we concatenate image tokens in $T$ frames with a learnable class token. The output of our visual encoder is $\mathbf{E_v} \in \mathbb{R}^D$. For the text encoder, we employ a 12-layer GPT-like Transformer (Radford et al., 2019) that input tokens after BPE tokenization (Sennrich, 2015). The output of our text encoder is $\mathbf{E_t} \in \mathbb{R}^D$.

**Motion Adapter.** Intuitively, to encode visual representations at the same level of detail as the languages, it is essential to utilize a greater number of frames as input. Since increasing the number of frames in training will result in unacceptable computational overhead, inspired by the PEFT technique in LLMs (Ding et al., 2023), we propose to use a lightweight motion adapter. The motion adapter is injected between the layers of the visual backbone, and is tailored to learn the finer-grained details with a high framerate. Since the hand and object motion forms a spatiotemporal pattern, unlike previous methods (Pan et al., 2022; Xing et al., 2024) that only focus on learning temporal information, our module is designed to learn both spatial and temporal information.

Our motion adapter is attached to the top of each of the $N$ transformer layers. Without loss of generality, here we illustrate the motion adapter for one transformer layer and illustrate it in Figure 4. Denote $\mathbf{Y} \in \mathbb{R}^{L \times D}$ as the output of a transformer layer in $\mathcal{F}_v$ where $L$ is the number of tokens, we first forward $\mathbf{Y}$ to a down-projection layer $W_{down}$ with ratio $\gamma$ and followed with a GELU activation function $\sigma$. Then, we use a 2D convolution layer `Conv2D` with kernel size $(k,k)$ to aggregate spatial information from each frame, followed by a 1D temporal convolution layer `TConv1D` and Linear layer $W_m$ to model the dynamics between adjacent frames. Finally, an up-projection layer $W_{up}$ is used to restore the dimension. Formally, the structure can be described as:

$$\mathbf{Y}' = \sigma(\mathbf{Y}W_{down}), \quad \mathbf{Y_s} = \mathrm{ReLU}(\mathrm{BN}(\mathrm{Conv2D}(\mathbf{Y}'))),$$
$$\mathbf{Y_{st}} = (\mathrm{TConv1D}(\mathbf{Y_s}))W_m, \quad \mathrm{MotionAdapter}(\mathbf{Y}) = \mathbf{Y} + \mathbf{Y_{st}}W_{up}, \tag{2}$$

where $W_{down} \in \mathbb{R}^{D \times \gamma D}$, $W_m \in \mathbb{R}^{\gamma D \times \gamma D}$ and $W_{up} \in \mathbb{R}^{\gamma D \times D}$. BN denotes BatchNorm2D.

**Co-training Strategy.** In EgoVideo, the motion adapter receives input at a higher framerate to capture the fine-grained information. Additionally, the backbone must be trained to fully adapt to the egocentric domain. Thus, different from previous PEFT methods that freeze the backbone and only train the adapter part, we need to train both the backbone and adapter parameters. Motivated by the architecture of (Feichtenhofer et al., 2019), we employ a co-training strategy to train the backbone and the motion adapter jointly.

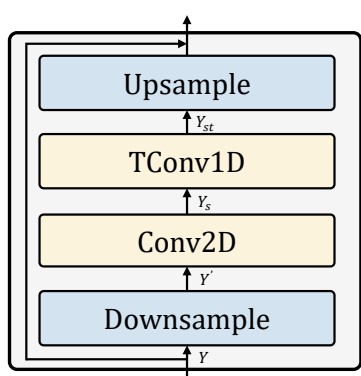

Figure 4: Architecture of our motion adapter. We use a 2D convolution layer and a 1D temporal convolution layer to capture the spatial and temporal dynamics efficiently.

Specifically, we use an upsampling parameter $\lambda$ to sample the input using two sampling rates. For the input $x_l \in \mathbb{R}^{T \times H \times W \times C}$ with a low sampling rate, we pass it through the backbone and unfreeze all parameters. As a result, we get the output $\mathbf{E_{vl}} \in \mathbb{R}^D$. For the input with a higher sampling rate $x_h \in \mathbb{R}^{\lambda T \times H \times W \times C}$, we pass the input through both the backbone and adapter parameters and get the output $\mathbf{E_{vh}} \in \mathbb{R}^D$, during which we freeze the parameters of the backbone and only train the adapter. Finally, for the outputs of the two pathways, we concatenate them and pass them through a fully connected layer to obtain the final output $\mathbf{E_v} \in \mathbb{R}^D$:

$$\mathbf{E_{vl}} = \mathcal{F}_{backbone}(x_l), \quad \mathbf{E_{vh}} = \mathcal{F}_v(x_h),$$
$$\mathbf{E_v} = [\mathbf{E_{vl}}; \mathbf{E_{vh}}]W_o, \tag{3}$$

Where "[;]" denotes the concatenation operation, $\mathcal{F}_{backbone}$ denotes the visual backbone, $\mathcal{F}_v$ denotes $\mathcal{F}_{backbone}$ with motion adapter and $W_o \in \mathbb{R}^{2D \times D}$. With this strategy, we integrate the training of the backbone and adapter into a single stage, reducing the cost of data and computation.

**Vision-Text Alignment.** We follow the standard InfoNCE (Oord et al., 2018) loss as the objective for alignment between visual embedding $\mathbf{E_v}$ and text embedding $\mathbf{E_t}$. For a sampled batch $\mathcal{B}$, we have:

$$\mathcal{L} = \frac{1}{|\mathcal{B}|} \sum_{(\mathbf{E_v^i}, \mathbf{E_t^i}) \in \mathcal{B}} \left( \log \frac{e^{s(\mathbf{E_v^i}, \mathbf{E_t^i})/\tau}}{\sum_{\mathbf{E_t^j} \in B} e^{s(\mathbf{E_v^i}, \mathbf{E_t^j})/\tau}} + \log \frac{e^{s(\mathbf{E_v^i}, \mathbf{E_t^i})/\tau}}{\sum_{\mathbf{E_v^k} \in B} e^{s(\mathbf{E_v^k}, \mathbf{E_t^i})/\tau}} \right), \tag{4}$$

Table 2: Zero-shot performance comparison on 4 tasks between methods with different model sizes ('B' for base, 'L' for large, and 'G' for our 1B parameter backbone). Our **EgoVideo** outperforms previous methods with less but higher-quality pretraining data. Specifically, our **EgoVideo-G** achieves significant performance improvements across all datasets.

| Method (ZS) | Data Size | EK-100 MIR | | EK-100 CLS | | EGTEA | | EgoMCQ | |
| --- | --- | --- | --- | --- | --- | --- | --- | --- | --- |
| | | *mAP* | *nDCG* | *Top1-acc.* | *Top5-acc.* | *Mean-acc.* | *Top1-acc.* | *Intra* | *Inter* |
| EgoVLPv2 | 4M | 26.7 | 29.1 | - | - | - | - | 60.9 | 91.0 |
| LaViLa-B | 35M | 30.9 | 32.0 | 16.4 | 34.4 | 28.9 | 35.4 | 59.9 | 93.8 |
| AVION-B | 35M | 32.9 | 32.7 | - | - | - | - | - | - |
| EMBED-B | 38.3M | 36.0 | **34.9** | 19.0 | 39.0 | 37.0 | 42.7 | 61.3 | 94.5 |
| **EgoVideo-B** | 7.4M | **36.5** | 34.5 | **22.4** | **43.3** | **43.6** | **51.0** | **64.6** | **95.0** |
| LaViLa-L | 35M | 36.1 | 34.6 | 20.8 | 41.4 | 34.1 | 40.1 | 63.1 | 94.5 |
| AVION-L | 35M | 37.6 | 35.3 | - | - | - | - | - | - |
| Helping Hands | 4M | 37.5 | **37.8** | - | - | 39.1 | 46.6 | 63.0 | 94.5 |
| EMBED-L | 38.3M | 40.8 | 37.5 | 22.8 | 45.0 | 40.3 | 46.7 | 64.7 | 95.6 |
| **EgoVideo-L** | 7.4M | **41.8** | 37.0 | **24.0** | **46.8** | **47.1** | **51.7** | **65.5** | **95.9** |
| **EgoVideo-G** | 7.4M | **47.1** | **39.0** | **28.5** | **54.3** | **58.0** | **63.0** | **69.1** | **96.6** |

where $s(\mathbf{E_v^i}, \mathbf{E_t^i})$ denotes dot product operation between the $i$-th sample in the batch of $\mathbf{E_v}$ and $\mathbf{E_t}$, and $\tau$ is a temperature parameter that scales the similarity scores.

# 4 EXPERIMENTS

## 4.1 DATASETS AND EVALUATION PROTOCOLS

**Pretraining Dataset.** As stated in the previous section, the source of our pretraining data comes from Ego4D (Grauman et al., 2022) and How2-Interlink-7M (Wang et al., 2024a). After processing by our HOD pipeline, the total amount of data is 7.4M clips.

**Evaluation Protocols.** We follow previous works (Zhao et al., 2023; Pramanick et al., 2023) and use the following evaluation protocols. (1) Zero-shot (ZS): the pretrained video-text encoders are directly applied to the downstream datasets to perform video-text retrieval tasks without any additional tuning. For classification, we compute the similarity score between the video clip and the textual descriptions of all possible classes. (2) Finetuned (FT): This approach involves taking the pretrained video-text model and performing end-to-end finetuning on the training split of the target downstream dataset. (3) Feature-based: We extract video features using a frozen encoder and only train a task-specific head on the downstream dataset.

**Model Architecture and Hyperparameters.** Our vision-language model follows the initialization of CLIP (Radford et al., 2021), which is composed of a vision encoder and a text encoder. For the base and large models, ViT is used as our vision encoder, and we use a temporal position embedding to learn temporal information, which is randomly initialized. For our giant size model, we use Internvideo2 (Wang et al., 2024b). For hyperparameters, we use $T = 4$ and $\lambda = 4$ for frame inputs, and we use downsample ratio $\gamma = 0.5$ for the motion adapter. During pretraining, we freeze the temperature parameter $\tau = 0.07$. More details are placed in Appendix C.

**Downstream Tasks.** We evaluate models on several egocentric downstream tasks: (1) Epic-Kitchens-100 (Damen et al., 2020) (EK-100) tasks. For this dataset, we evaluate our method on multi-instance retrieval (EK-100 MIR) and action recognition (EK-100 CLS) tasks; 2) Ego4D (Grauman et al., 2022) tasks. For Ego4D, we evaluate our model on multiple choice questions (EgoMCQ) (Li et al., 2021), and natural language query (EgoNLQ) and moment query (EgoMQ) tasks; 3) EGTEA (Li et al., 2018) tasks. We evaluate our model on the action recognition task that is focused on fine-grained cooking activities and hand-object interaction. 4) Other tasks. We also evaluate our model on GTEA (Fathi et al., 2011b) and HOI4D (Liu et al., 2022) datasets for the action segmentation task. Meanwhile, to show the generalization ability of our learned video representation, we evaluate the task success rate on Franka Kitchen dataset (Gupta et al., 2019), a simulation environment for embodied AI.

Table 3: Fine-tuning performance of models of different sizes on 5 tasks. Compared to the previous SOTA EMBED-L, our EgoVideo achieves new state-of-the-art performances on all datasets.

| Method (FT) | EK-100 MIR | | EK-100 CLS | EGTEA | EgoNLQ | EgoMQ | |
|---|---|---|---|---|---|---|---|
| | *mAP* | *nDCG* | *Top1-acc.* | *Top1-acc.* | *R1@0.5* | *R1@0.5* | *mAP* |
| EgoVLPv2-B | 47.3 | 61.9 | - | - | 7.9 | 31.1 | 12.2 |
| EgoVideo-B | **52.7** | **65.3** | **49.8** | **74.6** | **8.1** | **34.7** | **14.7** |
| Helping Hands-L | - | - | - | - | 7.9 | 33.4 | 16.0 |
| LaViLa-L | 50.9 | 66.5 | 51.0 | 76.0 | 7.3 | 32.5 | 13.4 |
| EMBED-L | 56.0 | 67.9 | 51.9 | 76.1 | 8.5 | 33.9 | 15.1 |
| EgoVideo-L | **56.6** | **69.0** | **53.7** | **79.2** | **8.9** | **36.6** | **17.1** |
| EgoVideo-G | **60.3** | **70.0** | **56.0** | **80.0** | **10.0** | **38.7** | **19.6** |

## 4.2 COMPARISON TO STATE-OF-THE-ART

**Zero-shot Evaluation.** Table 2 shows the results on 4 tasks in the zero-shot setting. We compare our method against previous egocentric representation learning methods including EgoVLPv2 (Pramanick et al., 2023), LaViLa (Zhao et al., 2023), AVION (Zhao & Krähenbühl, 2023), Helping Hands (Zhang et al., 2023) and EMBED (Dou et al., 2024). Notably, despite the use of refined captions and significantly larger training datasets, LaViLa, AVION, and EMBED fail to achieve results as our EgoVideo. In the following experiments we will demonstrate that both our high-quality HOD data and our design of the EgoVideo model play important roles in achieving good performance. Helping Hands uses a stronger backbone TimeSformer (Bertasius et al., 2021) and adds additional decoders for auxiliary object-oriented tasks. However, our method can still outperform Helping Hands, demonstrating the superiority of our representation learning scheme.

Specifically, in the EK-100 MIR task, our EgoVideo outperforms EMBED by 0.5%, 1.0%, 6.3% in mAP and significantly outperforms the LaViLa at the same model size. In the EK-100 CLS task, our EgoVideo-B model demonstrates superior performance with a top-1 accuracy of 22.4% and a top-5 accuracy of 43.3%, significantly outperforming LaViLa-B and EMBED-B.

On the EGTEA dataset known for its focus on hand-object interactions, our EgoVideo-B achieves a mean accuracy of 43.6% and a top-1 accuracy of 51.0%, surpassing EMBED-B and even EMBED-L. This underscores the importance of learning hand object dynamics and shows the strong generalization capability of our model. The EgoMCQ task further highlights the efficacy of our method, with EgoVideo-B outperforming LaViLa-B by 4.7%, 1.2% and EMBED-B's by 3.3% and 0.5% on the inter-class and intra-class accuracy, respectively. Our EgoVideo-L model also shows significant improvements with an inter-class accuracy of 65.5% and an intra-class accuracy of 95.9%. These results demonstrate the superior performance and generalization capability of our method without any additional supervision. We take a step forward to explore the scaling law in egocentric representation learning, finding that EgoVideo-G has elevated performance to the next level.

**Fine-tuning Evaluation.** Table 3 shows the result of the fine-tuning evaluation. Our EgoVideo method outperforms previous approaches across all tasks and datasets. Our EgoVideo-B demonstrates significant performance enhancements compared to EgoVLPv2-B, with improvements of 5.4% and 2.5% in mAP for the EK-100 MIR and EgoMCQ tasks, respectively. This performance is even comparable to the larger LaViLa-L. For our EgoVideo-L, we observe consistent improvements across all tasks, including a substantial enhancement by 1.8% and 3.1% in EK-100 CLS and EGTEA action recognition tasks, highlighting the superior performance of our model in fine-grained action understanding. Moreover, we achieve improvements of 0.4% in R1@0.5 in the EgoNLQ task and 2.7% and 2.0% in R1@0.5 and mAP in the EgoMQ task, confirming the richness of representations learned by our model and its capacity to capture intricate hand-object interaction information.

## 4.3 ABLATION STUDIES

**Pretraining Data.** We first conduct experiments by fixing the models and varying the pretraining data. Here we choose to use AVION for fair comparison since both AVION and EgoVideo use ViT as the backbone. As shown in Table 4, both our EgoVideo and AVION achieve the best performance

when the combination of Ego4D-HOD data and How2-HOD data is used, and EgoVideo consistently outperforms AVION when trained on the same data, emphasizing the effectiveness of the model design. Comparing models trained with EgoClip and Ego4D-HOD (rows 1,2 and 5,6), it is clear that significant improvements can be observed in the EK-100 MIR and EGTEA tasks. Adding additional data from How2-HOD can improve both models substantially (rows 1,3 and 5,7). Furthermore, when using only Ego4D-HOD, the performance on EGTEA surpasses EgoClip and How2-HOD together, indicating the beneficial impact of our data on fine-grained dynamics understanding.

Table 4: Ablations on different pretrain datasets, include original EgoClip (Zhao & Krähenbühl, 2023), Ego4D-HOD and How2-HOD selected by our classifer from How2Interlink-7M.

| ID | Model | Ego4D-EgoClip | Ego4D-HOD | How2-HOD | EK-100 MIR | | EGTEA | |
|---|---|---|---|---|---|---|---|---|
| | | | | | mAP | nDCG | Mean-acc. | Top1-acc. |
| 1 | | ✓ | | | 27.3 | 29.3 | 26.2 | 30.5 |
| 2 | AVION-B | | ✓ | | $31.0_{(+3.7)}$ | $31.3_{(+2.0)}$ | $32.3_{(+6.1)}$ | $37.0_{(+6.5)}$ |
| 3 | | ✓ | | ✓ | $33.2_{(+5.9)}$ | $32.5_{(+3.2)}$ | $31.6_{(+5.4)}$ | $35.6_{(+5.1)}$ |
| 4 | | | ✓ | ✓ | $\mathbf{34.4}_{(+7.1)}$ | $\mathbf{33.7}_{(+4.4)}$ | $\mathbf{39.4}_{(+13.2)}$ | $\mathbf{46.4}_{(+15.9)}$ |
| 5 | | ✓ | | | 31.1 | 32.0 | 30.8 | 36.0 |
| 6 | EgoVideo-B | | ✓ | | $34.4_{(+3.3)}$ | $33.9_{(+1.9)}$ | $41.1_{(+10.3)}$ | $47.9_{(+11.9)}$ |
| 7 | | ✓ | | ✓ | $35.5_{(+4.4)}$ | $34.1_{(+2.1)}$ | $40.8_{(+10.0)}$ | $47.1_{(+11.1)}$ |
| 8 | | | ✓ | ✓ | $\mathbf{36.5}_{(+5.4)}$ | $\mathbf{34.5}_{(+2.5)}$ | $\mathbf{43.6}_{(+12.8)}$ | $\mathbf{51.0}_{(+15.0)}$ |

Table 5: Comparison of the number of parameters.

| Method | Backbone | Params | EK100 mAP |
|---|---|---|---|
| LaViLa-B | TSF-B | 121M | 30.9 |
| AVION-B | ViT-B | 86M | 32.9 |
| EMBED-B | TSF-B | 121M | 36.0 |
| EgoVideo-B | ViT-B | 112M | **36.5** |
| LaViLa-L | TSF-L | 438M | 36.1 |
| AVION-L | ViT-L | 307M | 37.6 |
| EMBED-L | TSF-L | 438M | 40.8 |
| EgoVideo-L | ViT-L | 375M | **41.8** |
| EgoVideo-G | ViT-G | 1050M | **47.1** |

Table 6: The computational cost during inference. Views = #frames × #spatial crops × #temporal clips. "Extra GFLOPs" means extra computation compared to ViT.

| Method | Views | GFLOPs | Extra GFLOPs |
|---|---|---|---|
| ViT-B | $4 \times 1 \times 3$ | 201 | - |
| ViT-B | $16 \times 1 \times 3$ | 804 | - |
| LaViLa-B | $16 \times 1 \times 3$ | 1432 | |
| EgoVideo-B | $16 \times 1 \times 3$ | 1092 | 288 |
| ViT-L | $4 \times 1 \times 3$ | 1047 | - |
| ViT-L | $16 \times 1 \times 3$ | 4188 | - |
| LaViLa-L | $16 \times 1 \times 3$ | 4956 | |
| EgoVideo-L | $16 \times 1 \times 3$ | 5350 | 1162 |

**Model Size and Inference Computational Cost.** Table 5 compares the number of parameters. Our EgoVideo model maintains a relatively small parameter count, and even with the addition of the motion adapter, the total remains lower than that of LaViLa and EMBED, highlighting the efficiency of our approach. Meanwhile, in Table 6 we compare the inference computational cost of our EgoVideo with ViT and LaViLa. Thanks to our MotionAdapter, the increase in inference time for our model compared to ViT at 16 frames, is only similar to ViT's inference time at 4 frames.

**Training Efficiency.** In Table 7, we compare the performance and computational speed of our EgoVideo-B with AVION-B, where AVION-B is trained under two different parameter settings: pre-training with 16 frames and pre-training with 4 frames. Our EgoVideo is trained in a mixed 16 and 4 frame fashion, thus being faster than directly using all 16 frames to train the whole backbone. Meanwhile, EgoVideo achieves the best performance on the EK-100 MIR and EGTEA datasets. These results strongly demonstrate the effectiveness of our training strategy and motion adapter design in the EgoVideo model.

**Motion Adapter vs Other Adapters.** We compare our motion adapter with the standard Adapter (Houlsby et al., 2019) and the ST-adapter (Pan et al., 2022). In the standard adapter, we only use a downsample MLP and upsample MLP, while in the ST-adapter, we perform convolution operations solely along the temporal dimension. As shown in Table 8, the results on the EK-100 MIR

task demonstrate that both the ST-adapter and our motion adapter outperform the standard adapter. This improvement can be attributed to the limited parameters of the standard adapter, which restrict its ability to capture complex, fine-grained information. Compared to the ST-adapter, our Motion Adapter achieves the best performance by adding a spatial convolution operation, suggesting that both spatial and temporal information are crucial for egocentric video representation learning.

Table 7: Ablations on training strategy. Models are all trained on our Ego4D-HOD dataset with 10 epochs.

| Method | GPU Hours | EK-100 MIR | | EGTEA |
|--------|-----------|------------|------|-------|
| | | *mAP* | *nDCG* | *Top1-acc.* |
| AVION-4f | **95.5** | 34.4 | 33.7 | 46.4 |
| AVION-16f | 395.5 | 36.2 | 34.3 | 47.4 |
| EgoVideo | 180.6 | **36.5** | **34.5** | **51.0** |

Table 8: Experiment with different adapters. Our motion adapter achieves the best performance with a small increase in parameters.

| Design | Param Size | EK-100 MIR | |
|--------|-----------|------------|------|
| | | *mAP* | *nDCG* |
| Adapter | 8.28M | 34.7 | 33.0 |
| ST-adapter | 10.08M | 35.9 | 34.1 |
| Motion Adapter | 26.01M | **36.5** | **34.5** |

### 4.4 FEATURE-BASED EVALUATION ON OTHER TASKS

With the knowledge of hand-object dynamics, EgoVideo features can well generalize to other human behavior understanding tasks and robot manipulation tasks. Table 9 shows the results of Action Segmentation on the HOI4D (Liu et al., 2022) and GTEA (Fathi et al., 2011a) datasets, using features extracted from I3D (Carreira & Zisserman, 2017), AVION, and our EgoVideo. The results demonstrate our EgoVideo is also effective in the action segmentation task, especially for HOI4D which requires differentiating fine-grained hand-object interaction.

Also, we test the generalization capability of EgoVideo on the robot manipulation task on the Franka Kitchen dataset (Gupta et al., 2019). We follow the same setting and compare with previous robotic representation learning works MVP (Radosavovic et al., 2023), Voltron (Karamcheti et al., 2023) and MPI (Zeng et al., 2024). For MPI we compare both MPI with and without additional detection supervision. From Table 10, our EgoVideo can consistently surpass MVP and Voltron on the "Turn knob (TK)", "Open Microwave (OM)" and "Open door (OD)" tasks. While MPI uses additional detection and prediction transformers and performs better than EgoVideo on two tasks, EgoVideo still performs comparably in overall success rate. Complete results with more details and analyses can be seen in Appendix E. The results strongly prove the delicacy and generalization ability of our EgoVideo learned representations.

Table 9: Experiment on the action segmentation task. We report results of ASFormer (Yi et al., 2021) with different input features.

| Feature | HOI4D | | GTEA | |
|---------|-------|------|------|------|
| | *F1@50* | *Edit* | *F1@50* | *Edit* |
| I3D | 35.0 | 80.3 | 79.2 | 84.6 |
| AVION | 70.2 | 89.1 | 84.5 | 89.4 |
| EgoVideo | **74.8** | **90.1** | **87.1** | **90.1** |

Table 10: Results on Franka Kitchen. We report the success rate (%) on 50 sampled trajectories.

| Method | TK | OM | OD | Avg. |
|--------|------|------|------|------|
| MVP | 79.0 | 41.0 | 48.0 | 56.0 |
| Voltron | 76.0 | 41.0 | 45.3 | 54.1 |
| MPI | **85.5** | 49.0 | 52.5 | 62.3 |
| EgoVideo | 80.1 | **65.0** | **52.7** | **66.0** |
| MPI+Det | 89.0 | 54.0 | 57.7 | 66.9 |

## 5 CONCLUSION

In this work, we inject fine-grained hand-object dynamics into egocentric video representation learning. Our method addresses the drawbacks of the existing method from two perspectives. On the data side, we propose **HOD**, a novel framework to generate new paired video-language data, where the language contains intricately depicted hand-object dynamics. On the model side, we propose **EgoVideo**, where we use a model with a motion adapter combined with a co-training technique, to fully exploit the fine-grained dynamics provided by HOD data in the representation learning process. Experimental results demonstrate that our method achieves state-of-the-art performance across multiple downstream tasks, and can generalize in the embodied manipulation environment.

**Acknowledgement** This work is funded in part by the National Key R&D Program of China (2022ZD0160201), and Shanghai Artificial Intelligence Laboratory, and JSPS KAKENHI Grant Number JP22KF0119.

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

# A   DETAILS ABOUT HOD

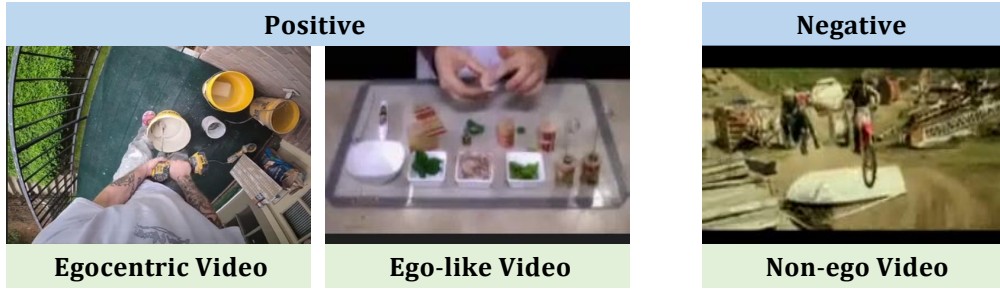

Figure 5: Examples of egocentric video/ego-like video and non-ego video.

**Data Selection** Since our HOD involves data from not only Ego4D but also How2link-7M, we use a style classifier $\mathcal{P}$ to filter egocentric style videos from the How2link-7M dataset. Specifically, our style classifier employs a simple two-layer MLP architecture. We utilize InternVideo2 (Wang et al., 2024b) to extract video features from all videos of the How2-Interlink7M dataset. After that, we manually annotate 10,000 clips with positive and negative labels, where the positive label indicates this video is an egocentric video (or ego-like video). Examples of positive and negative labeled videos can be found in Figure 5. We randomly select 10% of these clips to form the validation set. After training our classifier on the train set, we get 89% accuracy on the validation set.

**HOD Rephraser** We use Yi-34B model to generate hand object dynamics narrations. The Yi-34B model is trained on a corpus of over 150,000 high-quality texts and its model weights are open-source, which has a high ranking among all existing open-source Large Language Models. We directly use the model without finetuning.

To generate reliable narrations, we need to convert the obtained hand-object information into appropriate texts. For the movement trajectories of hands and objects, we directly calculate the center points of the bounding boxes and perform normalization to get a sequence $L = ((w_0, h_0), (w_1, h_1), ..., (w_{15}, h_{15}))$. To determine whether the object is contacted by the left hand or right hand separately, or contacted by both hands simultaneously, we use a generalized IoU function on the left-contact object and right-contact object. For IoU value greater than 0.9, we classify the object as being contacted by both hands.

Subsequently, we prompt the LLM with a system prompt of:

```
## System Prompt
Now you are a captioning assistant, you need to generate hand object interaction
caption and combine them with the origin narration. Given the origin narration
of the video clip and spatial localization ([x, y]) of hands and objects in the
clip, please help me describe the direction of motion of the left and right hands,
their relative relationship to objects and whether they are touching or not. Do
not
mention the pixel info. Two_hand_object means objects with two hands in contact,
left_hand_object means objects with left hand in contact, right_hand_object means
objects with right hand in contact.

## Hand Object Dynamics
left_hand:((w_0,h_0),(w_1,h_1),...,(w_15,h_15))
right_hand:((w_0,h_0),(w_1,h_1),...,(w_15,h_15))
left_hand_object:((w_0,h_0),(w_1,h_1),...,(w_15,h_15))
right_hand_object:((w_0,h_0),(w_1,h_1),...,(w_15,h_15))
two_hand_object:((w_0,h_0),(w_1,h_1),...,(w_15,h_15))
origin narration:  C takes a scissors.
```

```
## System Prompt
Please help me summarize the direction of movement of the left hand, right hand,
and objects, and generate a new caption based on the original caption.  It is
strictly forbidden to mention the frame number and spatial position coordinates in
the description.
```

For the computational cost, it takes around 2 days to extract bounding boxes from all vision-language clips and 3 days to generate narrations with LLM using 32 A100 GPUs, resulting in a total of around 4000 GPU·hours.

**Data Evaluation.**    Here, we provide a detailed explanation of our evaluation process. First, we prompted the LLM to generate new narrations with different verbs/nouns using the prompt:

```
Please help me modify the key verbs and nouns in this sentence to slightly alter
its meaning while keeping the sentence structure largely unchanged.  Just return
the modified sentence to me.  Ensure the semantic shift is minimal, such as
changing one or two verbs and nouns.
```

Then we utilized GPT-4o as a judge to determine the quality of the narrations with the prompt:

```
You are a judge.  There are 16 frames in the video, I have three captions and
need your help to score the three captions based on three criteria:  relevance,
accuracy, and level of detail.  The score ranges from 0 to 10, with a higher score
indicating better quality of the caption.  You can just answer me in the following
format:  First:  score1, Second:  score2, Third:  score3. First caption:  text1
Second caption:  text2 Third caption:  text3
```

As mentioned in the main manuscript, to further validate the quality of our HOD dataset, we utilize two standard unsupervised automatic metrics to evaluate the quality of narrations. We use the human narration as the ground truth and compare our HOD data with LaViLa-Narrator on METEOR and CIDEr scores. The results in Table 11 reveal that while our HOD data achieves a slightly lower METEOR score, it outperforms LaViLa-Narrator in CIDEr. This discrepancy arises because many LaViLa narrations closely mirror the original text, whereas our narrations incorporate additional dynamic information. Although our performance does not drastically exceed that of LaViLa, the results demonstrate that our narrations successfully retain the original semantic content.

Table 11: Comparison with LaViLa-Narrator on the narration quality.

| Text | METEOR | CIDEr |
|---|---|---|
| LaViLa-Narrator | 0.45 | 0.34 |
| HOD | 0.39 | 0.40 |

**Limitations and future work.**    Our model relies on the quality of hand-object detection and the rephrasing of LLM, which may include error accumulation. In addition to reducing the error in the data construction, exploring how the hand-object dynamics can be better involved into language or in other formats is a promising direction for our future work.

# B    DATASET DETAILS

**Ego4D** Ego4D (Grauman et al., 2022) contains 3,670 hours of egocentric videos with temporally dense narrations. Each narration has a timestamp and an associated free-form sentence. We follow previous works ((Zhao et al., 2023),(Lin et al., 2022)) to prepare the Ego4D dataset for vision-language pretraining. Specifically, we drop the narrations that either contain "#unsure"/"#Unsure" tags or are shorter than 4 words. This results in 4M video-text clip pairs.

**Howto-Interlink7M** Howto-Interlink7M (Wang et al., 2024a) contains 1M videos and 7M clips, which is part of the broader Howto100M dataset. Diverging from the original dataset, clips in Howto-Interlink7M have concise descriptions, and dense region captions and leverage GPT-4 to generate comprehensive summaries from detailed annotation. We use a classifier to select 3.3M vision-text pairs from the dataset.

**EpicKitchens-100** The Epic-Kitchens-100 (EK-100) dataset (Damen et al., 2020; 2018) contains 100 hours of egocentric cooking videos. Each clip is annotated with a start and end timestamp, a short textual narration, and a verb and noun class that the narration belongs to. The action class can also be uniquely determined by combining the verb and the noun. In EpicKitchens-MIR, we use Mean Average Precision and normalized Discounted Cumulative Gain (nDCG) as evaluation metrics. In EpicKitchens-CLS, we use top-1 action accuracy and top-5 action accuracy as evaluation metrics.

**EGTEA** EGTEA (Li et al., 2018) contains 28 hours of cooking activities from 86 unique sessions of 32 subjects. In zero-shot evaluation, we compute the similarity score between every video embedding and the 106 text embeddings, and take the text embedding with the highest similarity score as the predicted class. In fine-tuning evaluation, we finetune the video encoder for action classification. using the linear probing protocol.

**GTEA** The Georgia Tech Egocentric Activities (GTEA) dataset (Fathi et al., 2011b) consists of seven distinct types of everyday activities, including making a sandwich, preparing tea, and brewing coffee. Each of these activities is demonstrated by four different individuals, resulting in a total of 28 unique video recordings. Each video captures around 20 fine-grained action instances, such as "take bread" or "pour ketchup," all occurring within approximately one minute. This dataset provides a comprehensive look at egocentric perspectives, making it an invaluable resource for research in activity recognition and human-computer interaction.

**HOI4D** The HOI4D dataset (Liu et al., 2022) represents a significant advancement in the study of category-level human-object interaction, offering a large-scale 4D egocentric resource enriched with detailed annotations. Comprising 2.4 million RGB-D egocentric video frames across more than 4,000 sequences, the dataset captures interactions performed by nine participants with 800 unique object instances spanning 16 categories within 610 diverse indoor environments. To foster advancements in category-level human-object interaction, HOI4D introduces three benchmarking tasks: semantic segmentation of 4D dynamic point cloud sequences, category-level object pose tracking, and egocentric action segmentation involving a variety of interaction targets.

**Franka Kitchen** The Franka Kitchen dataset (Gupta et al., 2019) is a comprehensive resource designed to facilitate research in robotic manipulation and human-robot interaction within a kitchen environment. This dataset comprises a diverse collection of videos showcasing a humanoid robot, Franka Emika Panda, performing various cooking tasks. The setup features a Franka robot with 9 degrees of freedom positioned within a kitchen environment equipped with various common household items, including a microwave, kettle, overhead light, cabinets, and an oven. This environment is designed for multitask objectives, requiring the robot to interact with these items to achieve specific goal configurations.

## C IMPLEMENTATION DETAILS

**Pretraining Details** We pre-train on the video-narration pairs generated by our HOD from Ego4D and How-InterLink7M. We use AdamW optimizer with betas = (0.9,0.999) for 15 epochs. We use different settings for different size models. For EgoVideo-B, we adopt a batch size of 128 over 16 GPUs with a fixed learning rate of 5e-5, For EgoVideo-L, we use a batch size of 32 over 16 GPUs with a fixed learning rate of 3e-5. For EgoVideo-G, we choose to use a batch size of 16 over 16 GPUs with a fixed learning rate of 1e-5. For input frames, we preprocess the frames by resizing the shorter side to 320 pixels, which accelerates the data loading speed. Subsequently, we applied a standard RandomResizedCrop function (Zhao & Krähenbühl, 2023) with a scale parameter of (0.5, 1.0) to obtain the corresponding input frames.

**Finetuning Details** We finetune the downstream tasks using AdamW with $(\beta1, \beta2) = (0.9, 0.999)$ and weight decay of 0.05 with cosine annealing. Table 12 shows the hyperparameters details and in all tasks we use 8 GPUs for finetuning. During the training phase, we resize the shorter side of the video to 256 pixels and subsequently extract a 224×224 crop. During the testing phase, we scale the shorter side to 224 pixels and take a central 224×224 crop.

For the EgoNLQ task(Grauman et al., 2022), we build on the methodologies introduced by EgoVLP (Lin et al., 2022) and LAVILA (Zhao et al., 2023) for fairness. We adopt VSLNet (Zhang et al., 2020) as the task head. We train the task head for 50 epochs, using a learning rate of 3e-3, dropout

Table 12: Hyperparameters for Different Downstream Tasks

| Task | Model Size | Epochs | LR_start | LR_end | Batch Size |
|------|-----------|--------|----------|--------|------------|
| EK100-MIR | EgoVideo-B | 100 | 1e-6 | 1e-5 | 256 |
| | EgoVideo-L | 70 | 5e-7 | 5e-6 | 64 |
| | EgoVideo-G | 50 | 4e-7 | 4e-6 | 32 |
| EK100-CLS | EgoVideo-B | 100 | 1e-6 | 1e-5 | 256 |
| | EgoVideo-L | 70 | 5e-7 | 5e-6 | 64 |
| | EgoVideo-G | 60 | 4e-7 | 4e-6 | 32 |
| EGTEA | EgoVideo-B | 100 | 1e-6 | 1e-5 | 256 |
| | EgoVideo-L | 70 | 7e-7 | 7e-6 | 64 |
| | EgoVideo-G | 50 | 4e-7 | 4e-6 | 32 |

0.3, batch size 32 on a single A100 GPU. For the EgoMQ task, we use VSGN (Zhao et al., 2021) as our task head for training. We set batch size as 16, learning rate as 2e-4, gamma as 0.6, and train the task head on a single A100 GPU.

Table 13: Comparison of performance across different model sizes and vision-language datasets.

| Model | Pretrain Data | Data Size | EK-100 MIR | |
|-------|---------------|-----------|------|------|
| | | | mAP | nDCG |
| EgoVideo-B | EgoClip | 4M | 31.1 | 32.0 |
| EgoVideo-B | Ego4D-HOD | 4M | 34.4 | 33.9 |
| EgoVideo-L | EgoClip | 4M | 35.3 | 34.6 |
| EgoVideo-L | Ego4D-HOD | 4M | 38.3 | 35.9 |
| EgoVideo-G | EgoClip | 4M | 42.1 | 37.5 |
| EgoVideo-G | Ego4D-HOD | 4M | 44.8 | 38.2 |

# D  ADDITIONAL ABLATIONS

**Pretraining Data** To further demonstrate the effectiveness of our HOD, we fix the amount of data and conduct experiments using different model sizes. As shown in Table 13, with the same model size and the same data size, using our Ego4D-HOD data can consistently achieve improvement. Since one sample in EgoClip corresponds strictly to on sample in Ego4D-HOD, this table strongly demonstrates the high quality of our HOD data.

**Adapter Downsampling Ratio** We test the design of our motion adapter by studying the effect of adapter downsampling ratio $\gamma$, and show the result in Figure 6. It can be observed that as the value of $\gamma$ increases, the model's performance continues to improve. This indicates that our generated narrations contain rich semantic information and further validates the effectiveness of our motion adapter. To reduce computational overhead, we ultimately decide to set $\gamma = 0.5$.

**Frame number** We further study the effect of the number of sampled frames as input. We consistently use 4 frames as the sampling rate for the backbone part. The results in Table 14 indicate that as the number of frames increases from 4 to 16, the model's performance improves continuously from 34.2% to 36.5%. However, when the frame count reaches to 32, the performance plateaus, showing no significant improvement with further increases in frame count. Besides, increasing the number of frames beyond this point incurs substantial computational cost. As a result, we choose to use $\lambda = 4$ as the default value in our EgoVideo, balancing the speed and accuracy.

| Frame count | $\lambda$ | EK MIR mAP |
|:---:|:---:|:---:|
| 4 | 1 | 34.2 |
| 8 | 2 | 35.3 |
| 12 | 3 | 35.9 |
| 16 | 4 | **36.5** |
| 32 | 8 | **36.5** |

Table 14: Comparison of performance across different frame upsampling rate $\lambda$.

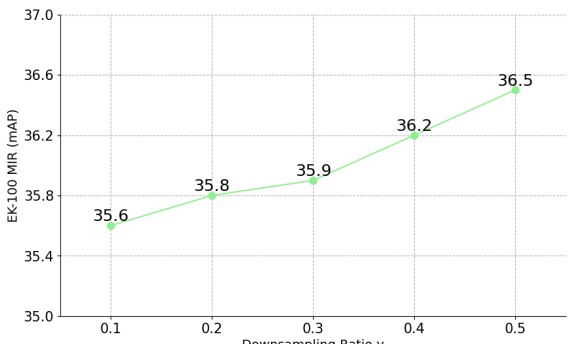

Figure 6: Ablations on Downsampling Ratio $\gamma$.

## E   ADDITIONAL RESULTS

Table 15: Comparison with the state-of-the-art methods on 50Salads, GTEA and HOI4D dataset.

| Feature | Method | GTEA | | | HOI4D | | |
|---|---|---|---|---|---|---|---|
| | | *F1@10, 25, 50* | *Edit* | *Acc* | *F1@10, 25, 50* | *Edit* | *Acc* |
| I3D | MS-TCN | 85.8 / 83.4 / 69.8 | 79.0 | 76.3 | 55.6 / 47.8 / 31.8 | 74.7 | 44.2 |
| I3D | MS-TCN++ | 88.8 / 85.7 / 76.0 | 83.5 | 80.1 | 54.7 / 46.5 / 30.3 | 75.2 | 42.2 |
| I3D | ASFormer | 90.1 / 88.8 / 79.2 | 84.6 | 79.7 | - | | |
| I3D | DiffAct | 92.5 / 91.5 / 84.7 | 89.6 | 82.2 | - | - | - |
| AVION | ASFormer | 92.5 / 91.0 / 84.5 | 89.4 | 81.4 | 84.4 / 81.1 / 70.2 | 89.2 | 74.2 |
| EgoVideo | ASFormer | **92.7 / 92.2 / 87.1** | **90.1** | **82.7** | **88.9 / 85.3 / 74.8** | **90.1** | **76.2** |

### E.1   DETAILS ON ACTION SEGMENTATION TASKS.

Action segmentation tests the representation on its understanding of the temporal dependencies of the video Huang et al. (2020b); Yi et al. (2021). We evaluate our model on two benchmark datasets: GTEA (Fathi et al., 2011b), and HOI4D (Liu et al., 2022). We follow the previous work to use four-fold cross-validations on both datasets. We use accuracy (Acc), the edit distance (Edit), and the F1 scores at overlap thresholds 10%, 25%, 50% (F1@10, 25, 50) as metrics for evaluation.

We use ASFormer (Yi et al., 2021) as the task head, with input features extracted by our Egovideo, I3D (Carreira & Zisserman, 2017), and AVION (Zhao & Krähenbühl, 2023). We follow (Chen et al., 2024), using learning rate = 5e-4, drop rate = 0.3, epoch = 100 for training. Table 15 presents the experimental results of our method and other recent approaches, including MS-TCN (Farha & Gall, 2019), MS-TCN++ (Li et al., 2020), ASFormer (Yi et al., 2021), and DiffAct (Saha et al., 2021). The results clearly show the high quality of our EgoVideo feature. With the same task head ASFormer, the EgoVideo feature can outperform the AVION feature consistently. EgoVideo can even help ASFormer to beat the stronger task head DiffAct.

### E.2   DETAILS ON FRANKA KITCHEN DATASET.

Here we introduce the details of experiments on the Franka Kitchen dataset(Gupta et al., 2019). In this dataset, we adopt 3 tasks, including "Turn the stove top knob (TK)", "Open the microwave (OM)" and "Open the left door (OD)". The goal is to predict 9-DoF joint velocities (7 joints and 2 grippers) based on the visual representations and proprioceptive states (*i.e.*, joint velocities). We follow the MPI mode (Zeng et al., 2024), which trains a shallow MLP policy network. For evaluation, we follow the R3M method (Nair et al., 2022) and (Karamcheti et al., 2023) and calculate the average success rates for each setting across the 3 tasks.

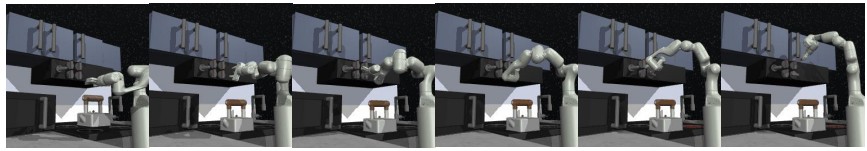

Turn on the knob

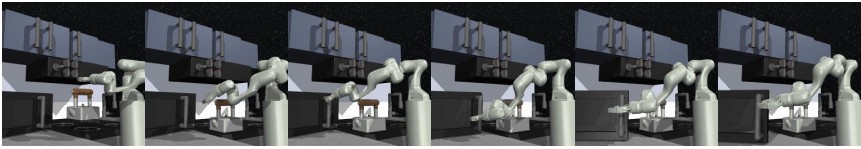

Open microwave

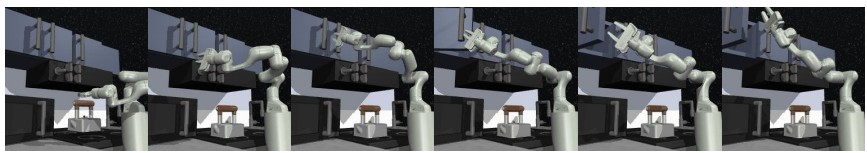

Open door

Figure 7: Qualitative results on the Franka Kitchen dataset. We show the tasks of turning the stovetop knob, opening the microwave and opening the door. All tasks are trained with 25 demonstrations.

We compare our EgoVideo with MVP (Radosavovic et al., 2023), Voltron (Karamcheti et al., 2023) and MPI (Zeng et al., 2024). MVP learns representation for robot manipulation by masked image modeling. Voltron takes a step forward to combine masked image modeling with vision text alignment. MPI designs detection and prediction transformers to use object detection signals as additional guidance. Notably, these works also use Ego4D as training data.

The experimental results in Table 10 indicate that our model significantly outperforms both the MVP and Voltron models by more than 10% in average success rate, and exhibits performance comparable to the more advanced MPI model, which integrates multiple pre-training tasks related to robot learning. When the MPI model is solely trained using contrastive learning and masked signal modeling as supervision, we achieve 3.7% improvements in average success rate than MPI model. When MPI incorporates the video prediction task, which has been proven crucial for robot learning, our average success rate is only 0.9% lower. This demonstrates the robust generalization capabilities of our model and highlights the contribution of our hand-object dynamics learning scheme to fine-grained hand operations. Figure 7 shows the qualitative results on turning on the knob, and opening the microwave and opening the door tasks.

## F    QUALITATIVE RESULTS

In Figure 8, we show more examples to compare narrations generated by our HOD with LaViLa Rephraser and the original narrations. We can observe that the narrations generated by our HOD model can well describe the hand-object dynamics (*e.g.*, 'The left hand moves downwards to touch the bicycle tire'). Moreover, compared to the LaViLa rephraser, which often merely changes word order or modifies nouns/verbs, our model can combine original actions to generate more semantically rich descriptions of actions and scenes, resulting in significantly higher quality narrations. (See the first example: our HOD generates 'Person C picks a card with their right hand, which is then handed to their left hand.' while LaViLa yields '#C C chooses a card/#C C selects a card/#C C picks a card').

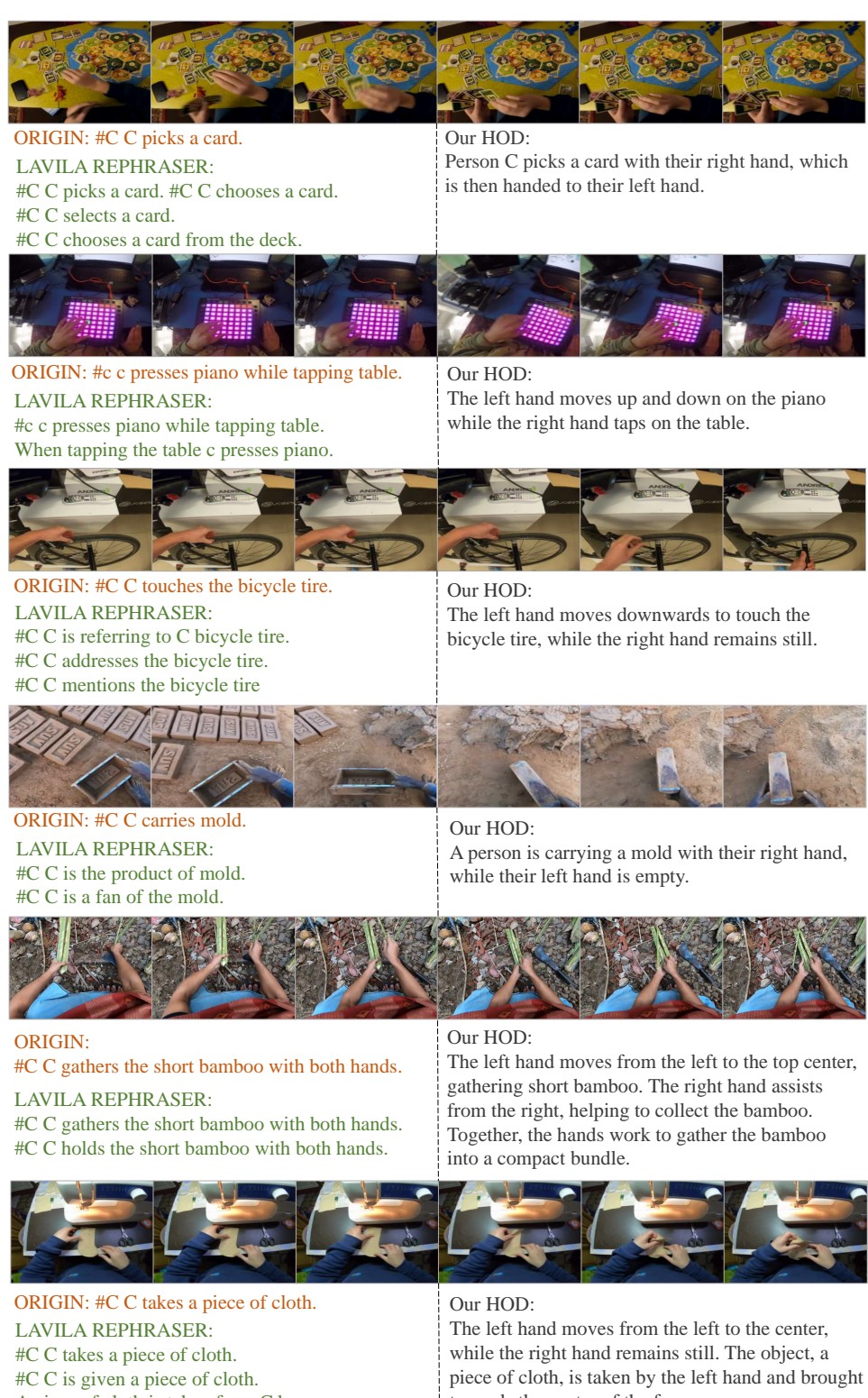

ORIGIN: #C C picks a card.
LAVILA REPHRASER:
#C C picks a card. #C C chooses a card.
#C C selects a card.
#C C chooses a card from the deck.

Our HOD:
Person C picks a card with their right hand, which is then handed to their left hand.

ORIGIN: #c c presses piano while tapping table.
LAVILA REPHRASER:
#c c presses piano while tapping table.
When tapping the table c presses piano.

Our HOD:
The left hand moves up and down on the piano while the right hand taps on the table.

ORIGIN: #C C touches the bicycle tire.
LAVILA REPHRASER:
#C C is referring to C bicycle tire.
#C C addresses the bicycle tire.
#C C mentions the bicycle tire

Our HOD:
The left hand moves downwards to touch the bicycle tire, while the right hand remains still.

ORIGIN: #C C carries mold.
LAVILA REPHRASER:
#C C is the product of mold.
#C C is a fan of the mold.

Our HOD:
A person is carrying a mold with their right hand, while their left hand is empty.

ORIGIN:
#C C gathers the short bamboo with both hands.
LAVILA REPHRASER:
#C C gathers the short bamboo with both hands.
#C C holds the short bamboo with both hands.

Our HOD:
The left hand moves from the left to the top center, gathering short bamboo. The right hand assists from the right, helping to collect the bamboo. Together, the hands work to gather the bamboo into a compact bundle.

ORIGIN: #C C takes a piece of cloth.
LAVILA REPHRASER:
#C C takes a piece of cloth.
#C C is given a piece of cloth.
A piece of cloth is taken from C bag.

Our HOD:
The left hand moves from the left to the center, while the right hand remains still. The object, a piece of cloth, is taken by the left hand and brought towards the center of the frame.

Figure 8: Comparison between: 1) the original Ego4D narrations; 2) LaViLa narrations; and 3) Narrations generated by our HOD. Our narration can describe the dynamic motion information of hands and objects, enhancing the semantic richness of the original narration.

