# OpenReview forum: "Modeling Fine-Grained Hand-Object Dynamics for Egocentric Video Representation Learning"
_ICLR.cc/2025/Conference — ICLR 2025 Poster_

### Official Review · Reviewer_PuQN · 2024-10-22

**Soundness:** 3
**Presentation:** 3
**Contribution:** 3
**Rating:** 6
**Confidence:** 4

**Summary:**

The authors introduce a new video-language model that learns video representations from fine-grained hand-object interaction descriptions. Unlike previous works that learn from abstract short texts with a fixed number of frames, they propose HOD and EgoVideo. HOD generates detailed descriptions by prompting a large language model (LLM) using noisy HOI bounding boxes, while EgoVideo incorporates a dual-branch video architecture that adopts a backbone and a lightweight adapter to handle inputs with different frame sampling rates. The authors conduct comprehensive experiments across various benchmarks to demonstrate the strength of their pretrained model.

**Strengths:**

- **Good Paper Writing**: The paper is well-organized, clear, and easy to follow.
- **Useful Architecture for Egocentric Video Understanding**: The architecture effectively improves performance in the egocentric domain, though the core idea of learning from varying frame sampling rates originates from SlowFast. The proposal of using the lightweight adapter increase only a few parameters.
- **Extensive Experiments and SoTA Performance**: The proposed EgoVideo-G (their largest model) achieves state-of-the-art performance across all benchmarks. The experiments cover a wide range of datasets and domains, including commonly used evaluation datasets in egocentric contexts like Epic-Kitchens-MIR and EgoMCQ, as well as evaluations in robotic manipulation. I appreciate the additional focus on robotic applications.

**Weaknesses:**

- **Lack of Analyses on HOD Data Corpus or Quality**: This paper lacks thorough analyses of the data corpus, especially considering the data generation pipeline is simple and naive. From the motivation of HOD, the authors claim to create 'HOI dynamics' using an HOI detector. However, since the HOI detector is image-based, there is no guarantee that the generated texts will capture temporal information. While I acknowledge the robustness of the HOI detector, I am unsure if this robustness ensures consistent temporal information and how much noise would impact the final model. From the data corpus, the authors do not analyze the data statistics of the generated texts at any perspective except for the experimental results. As a result, I have limited insight into the quality of the pretraining dataset. To illustrate the significance, take what LaViLa has done as an example. To demonstrate the effectiveness of generated sentences, LaViLa's Narrator evaluates their model on the validation set of Ego4D. I think similar experiments are essential to validate the usefulness of HOD. For instance, it is important to determine whether the original information is remained in the generated text and whether there are unreliable hallucinations present from Yi-34B.
- **Inadequate Analyses at Inference Time**: While mainstream video-language models use 4 frames during inference, EgoVideo requires sampling both 4 and 16 frames, leading to slower inference speeds and also potential unfair comparisons. Although Table 4 demonstrates the effectiveness of training EgoVideo jointly with 4 and 16 frames, another straightforward baseline of other models incorporating test-time augmentation, e.g., simply aggregating outputs of AVION-4f from both 4 and 16 frames, would enhance the evaluation. Besides, I think the comparison on GFLOPS during inference between models should also be provided.

**Questions:**

- **Lack of Citation**: The key idea in EgoVideo is closely related to the well-known SlowFast Network [SlowFastNetworks for Video Recognition](https://arxiv.org/abs/1812.03982) (ICCV 2019), but the authors do not cite it. Although their implementation ways are different, I recommend including this citation and adding a proper discuss.
- **Lack of Parameter Comparisons in Table 1**: In addition to dataset sizes, I suggest adding information on trainable parameters, as models like Helping Hands have significantly fewer trainable parameters than other methods.
- **Improper Comparison of EgoVideo-G**: One problem is the unfair comparisons involving EgoVideo-G with other models. While the Giant model demonstrates impressive performance, it lacks a fair comparison with other models that primarily use smaller variants. Besides, other backbones mainly leverage CLIP for initialization, whereas EgoVideo-G is initialized with the state-of-the-art InterVideo2 (L313). This not only means a larger model size but also a different initialization method. There is a lack of a straightforward ablation study or adequate explanation the choices.
- **Improper Comparison in Robot Manipulation**: While I appreciate the emphasis on robot manipulation, the comparison is unfair, as EgoVideo-G (2B) is significantly larger than other methods (e.g., MPI-Base, which is only 86M). The authors should include control speed and parameter sizes, as EgoVideo-G is likely slower and larger than the other methods, which impacts the evaluation.
- **Input to the Manipulation Model**: During the manipulation process, EgoVideo relies on multiple frames as input, whereas methods like MPI use only a single observed frame. I'm curious about how the authors address this difference.

---

> ### Author Response · Authors · 2024-11-21
> **Response to Reviewer PuQN**
>
> **Analysis of HOD data.** We thank the reviewer for the suggestion and provide the following analyses.
>
> 1. We analyze the distribution of the words of our HOD against the original EgoClip narrations. We select the top-30 most frequent words, and show their normalized frequency in Figure 3 in our revised manuscript. From this figure, we observe that words in the EgoClip narrations are more long-tailed, while our HOD captions show better distribution. Also, from the words, we can see a lot of "dynamic" words in our HOD data like "up","downwards" and "moves". This from one angle proves our HOD data is dynamically descriptive and of high quality.
> 2. We compare our HOD data with another data generation technique: LaViLa-Narrator[1]. We evaluate the CIDEr[2] and METEOR metric using the original human narration as the ground truth. From the table below, we can observe that compared with LaViLa-narrator, our CIDEr score is higher, while the METEOR score is slightly lower. This indicates that our narrations are semantically close to the original captions, preserving the original information. The lower METEOR score, we believe, is due to our narrations incorporating more fine-grained dynamic information, which is not present in the original captions.
>
>   | Text  | METEOR | CIDEr |
>   | --------------- | ------ | ----- |
>   | LaViLa-Narrator | **0.45**   | 0.34  |
>   | HOD(ours)       | 0.39   | **0.40**  |
>
>   3. To further verify the quality of our HOD data, we employ GPT-4o[3] for quality assessment. We randomly select 1000 clips(due to time and cost) and let GPT evaluate the score of the caption data for the video clip in a range from 0 to 10. To ensure GPT does not simply assign high scores based on the length of the captions, we also conduct random gerund replacements on our data for comparison. The results, summarized in the table below (also Table 1 in our submisstion), show that our HOD data have a significantly better GPT-Score.
>
> | Data            | GPT-Score |
> | --------------- | ------ |
> | EgoClip | 5.53   |
> | HOD-random      | 3.70  |
> | HOD      | **7.71**   |
>
> We hope these experiments and explanations can address the reviewers concern.
>
> **Inference Time.** We believe there is a misunderstanding. All the methods we compared (AVION[4], LaViLa, EMBED[5]) use 16 frames during inference, so our comparisons are fair. We evaluate the straightforward baseline by simply aggregating outputs of AVION-4f and AVION-16f, and we find that the results are similar to AVION-16f, which may be due to the use of the same model parameters. And we evaluate the GFLOPs results below.
>
> | Method     | Views  | GFLOPs | Extra GFLOPs |
> | ---------- | ------ | ------ | ------------ |
> | ViT-B      | 4×1×3  | 201    | -            |
> | ViT-B      | 16×1×3 | 804    | -            |
> | LaViLa-B      | 16×1×3 | 1432    | -            |
> | EgoVideo-B | 16×1×3 | 1092   | 288          |
> | ViT-L      | 4×1×3  | 1047   | -            |
> | ViT-L      | 16×1×3 | 4146   | -            |
> | LaViLa-L     | 16×1×3 | 4956   | -            |
> | EgoVideo-L | 16×1×3 | 5308   | 1162         |
>
> In the table we can observe that the increase in inference time for our model compared to ViT at 16 frames, is only similar to ViT's inference time at 4 frames, proving the efficientness of our motion adapter.
>
> [1]Zhao Y, Misra I, Krähenbühl P, et al. Learning video representations from large language models[C]//Proceedings of the IEEE/CVF Conference on Computer Vision and Pattern Recognition. 2023: 6586-6597.
>
> [2]Maluuba. nlg-eval. https://github.com/Maluuba/nlg-eval. Accessed: 2022-06-01.
>
> [3]Achiam J, Adler S, Agarwal S, et al. Gpt-4 technical report[J]. arXiv preprint arXiv:2303.08774, 2023.
>
> [4]Zhao Y, Krähenbühl P. Training a large video model on a single machine in a day[J]. arXiv preprint arXiv:2309.16669, 2023.
>
> [5]Dou Z Y, Yang X, Nagarajan T, et al. Unlocking exocentric video-language data for egocentric video representation learning[J]. arXiv preprint arXiv:2408.03567, 2024.

---

> ### Author Response · Authors · 2024-11-21
> **Response to Reviewer PuQN for Questions**
>
> **Slowfast network citation.** We appreciate the reviewer for the suggestion. We have added a discussion and cited SlowFast Network[1] in our revised manuscript.
>
> **Trainable parameters comparisons.** The table below shows the trainable parameters of different methods. Helping hands[2] uses frozen backbone initialized from LaViLa and uses only one additional Transformer Decoder for training, so it has significantly fewer trainable parameters.
>
> | Method     | Backbone  | Params | EK100-mAP |
> | ---------- | ------ | ------ | ------------ |
> | LaViLa-B      |TSF-B  | 121M    | 30.9            |
> | AVION-B      | ViT-B | 86M    | 32.9           |
> | EgoVideo-B | ViT-B| 112M  | 36.5       |
> | LaViLa-L      | TSF-L | 438M   | 36.1            |
> | AVION-L     | ViT-B| 307M  | 37.6         |
> | Helping Hands-L | TSF-L | 35M| 37.5 |
> | EgoVideo-L | ViT-L | 375M   | 41.8        |
> | EgoVideo-G | ViT-G | 1050M   | 47.1         |
>
> **Comparison of EgoVideo-G.** We compared the performance of models of the same size. The experimental results indicate that our EgoVideo-B and EgoVideo-L models achieve better performance compared to methods of the same size. Both our base and large models were initialized using CLIP. For EgoVideo-G, we aimed to explore the scaling law and test how effectiveness of our data can be on larger models. Therefore, we employed InternVideo2[3], which has a stronger temporal understanding capability, as the backbone. This led to a significant performance improvement in our model. We further examined the impact of our data on EgoVideo-G, as shown in the table below. This result proves that our HOD data can be better leveraged by larger models, showing the possible future directions of incorporating foundation models into egocentric video understanding.
>
> |Data|	EK100-MIR(mAP)|	EK100-MIR(nDCG)|
> | ---------- | ------ | ------ |
> |EgoClip	|43.8	|37.1|
> |HOD|	47.1|	39|
>
> **Comparison in Robot Manipulation.** In the robot manipulation task, we use EgoVideo-B to test for fairness, and we have clarified this issue in the revised manuscript. The EgoVideo-B model is on the same parameter scale as other baselines.
>
> **Input to the Manipulation Model.** Our model can dynamically adjust different number of input frames and we use just a single observed frame as input. The capability for single-frame inference underscores the robustness of our EgoVideo. This result also shows that using video-based pretraining can enhance the robot manipulation performance, which we believe reveals the potential for future works on leveraging the strength of video models into the robot manipulation tasks.
>
> [1]Feichtenhofer C, Fan H, Malik J, et al. Slowfast networks for video recognition[C]//Proceedings of the IEEE/CVF international conference on computer vision. 2019: 6202-6211.
>
> [2]Zhang C, Gupta A, Zisserman A. Helping hands: An object-aware ego-centric video recognition model[C]//Proceedings of the IEEE/CVF International Conference on Computer Vision. 2023: 13901-13912.
>
> [3]Wang Y, Li K, Li X, et al. Internvideo2: Scaling video foundation models for multimodal video understanding[J]. arXiv preprint arXiv:2403.15377, 2024.

---

> > ### Comment · Reviewer_PuQN · 2024-11-26
> >
> > Thank the authors for their responses. I have another question regarding robot manipulation task, particularly since EgoVideo-B is used for manipulations.
> >
> > **Question on Manipulation Tasks**
> >
> > While current works have shown that pretraining with only text objectives performs poorly (as evidenced by CLIP [1] in the MPI paper [2]), significant improvements are observed when visual-only pretraining objectives are employed. For example, mask image / video modeling (as used in Voltron [3] and MVP [4]) and time contrastive learning (as used in R3M [5]) have demonstrated notable performance gains.
> >
> > This disparity might be attributed to the strong awareness of object relations and spatial positions that visual-only pretraining objectives impart to the model.
> > On the other hand, EgoVideo's video-text pretraining objectives, being similar to those in CLIP, might have led to closer performance with CLIP.
> > This hypothesis might also be supported by the ablation experiments in Table 4 of Voltron, where the exclusion of the R-M3AE objective and reliance on language-only objectives caused a more significant performance drop.
> >
> >
> > To address this, I am curious whether the slowfast architecture or the HOD dataset contributes most to the gains in the robot manipulation task. I'd like to question the effect of HOD data, which also belongs to the CLIP paradigm. Also, since EgoVideo only uses one frame during inference, the slowfast architecture seems useless. Additional ablation studies in Table 4 might help clarify this question.
> >
> > Given the limited time for discussion, the authors could also share potential reasons or relevant findings from similar literature to further support the significant improvements in robot manipulation tasks.
> >
> > **Additional Comment**
> >
> > Besides, I do not admire the radar graph on the left side of Figure 1, where the pretraining data (inherited from the initialization of InternVideo2) and the model size (2B) of EgoVideo-G is significantly larger than previous SOTA, as I stated in question 3.
> > A more appropriate approach for visualizing Figure 1 would be to include EgoVideo-B and EgoVideo-L in the comparison.
> >
> > **References**
> >
> > [1] Learning transferable visual models from natural language supervision, Radford et al., ICML2021
> >
> > [2] Learning Manipulation by Predicting Interaction, Zeng et al., RSS2024
> >
> > [3] Language-driven representation learning for robotics, Karamcheti et al., RSS2023
> >
> > [4] Real-World Robot Learning with Masked Visual Pre-training, Radosavovic et al., PMLR2023
> >
> > [5] R3M: A Universal Visual Representation for Robot Manipulation, Nair et al, CoRL2022

---

> > > ### Author Response · Authors · 2024-11-26
> > > **Response to Reviewer PuQN**
> > >
> > > **Discussion on Pretraining for Manipulation Tasks.**
> > >
> > > We sincerely appreciate the reviewers' insightful comments. First, we would like to provide our thoughts on the CLIP-based/MAE-based pretraining objectives. As the reviewer observes, CLIP[1] performs poorly in the MPI[2] paper. We believe this is due to its pre-training on general image data, without emphasizing egocentric video and robotic dataset, both of which are more relevant to manipulation tasks. In contrast, methods like Voltron[3], MVP[4], and MPI benefit from pretraining on the Ego4D dataset. Regardless of whether they employ CLIP-based or MAE-based objectives, these methods achieve superior performance due to the domain-relevant data.
> > >
> > > While we agree that both masked image/video modeling and vision-language contrastive learning objectives can be beneficial for manipulation tasks, we hypothesize that the pretraining dataset itself plays a pivotal role in achieving strong performance. This aligns with our observation that MPI's superior results are driven by its use of interaction objects as supervision, emphasizing the importance of hand-object interactions in manipulation tasks.
> > >
> > > This hypothesis also explains why our HOD data, pre-trained with a simple CLIP-based objective, achieves competitive results—it leverages hand-object dynamics to provide targeted supervision.
> > >
> > > Due to time constraints, we were unable to explore the inclusion of masked video modeling in our pretraining objective. However, we plan to integrate this in future versions, as it has the potential to complement our existing approach and further enhance performance.
> > >
> > > **Which contributes more? HOD dataset or slowfast[5] architecture?**
> > >
> > > We believe that the **HOD dataset** plays a key role in enhancing manipulation tasks, as supported by the following points:
> > >
> > > 1.**Relevance of Spatial Reasoning and Object Understanding.** Previous works, such as OpenVLA[6] and MPI, demonstrate the importance of spatial reasoning and object detection for robot control and manipulation tasks. Our HOD data includes detailed descriptions of hand and object spatial positions, which are crucial for effective robot learning.
> > >
> > > 2.**Additional Baselines for Validation.** To further validate, as the reviewer suggested, we included two new baselines in our experiments:
> > >
> > > -  EgoVideo (b) + Ego4D, which uses the Ego4D dataset for pretraining but excludes the SlowFast architecture.
> > > -  EgoVideo + Ego4D, which incorporates the SlowFast architecture but does not use our HOD dataset.
> > >
> > > | Row ID | Method              | TK   | OM   | OD   | Avg. |
> > > | ------ | ------------------- | ---- | ---- | ---- | ---- |
> > > | 1      | CLIP                | 26.3 | 24.7 | 13.0 | 21.3 |
> > > | 2      | R3M[7]                 | 53.3 | 59.3 | 50.7 | 54.4 |
> > > | 3      | MVP                 | 79.0 | 41.0 | 48.0 | 56.0 |
> > > | 4      | Voltron             | 76.0 | 41.0 | 45.3 | 54.1 |
> > > | 5      | MPI                 | 85.5 | 49.0 | 52.5 | 62.3 |
> > > | 6      | EgoVideo(b) + Ego4D | 70.0 | 60.0 | 44.0 | 58.0 |
> > > | 7      | EgoVideo + Ego4D    | 72.0 | 56.5 | 48.3 | 58.9 |
> > > | 8      | EgoVideo + HOD      | 80.1 | 65.0 | 52.7 | 66.0 |
> > >
> > > The results are summarized as follows:
> > >
> > > - Comparing rows 6 (Baseline a) and 7 (Baseline b) in the table, the SlowFast architecture contributes slightly to performance improvement.
> > > - Comparing rows 7 (Baseline b) and 8 (ours), it is evident that the HOD dataset provides a much more significant contribution.
> > >
> > > We believe this is reasonable, since the slowfast architecture is **mainly designed for egocentric video understanding tasks** such as action recognition. The slight increase in robot manipulation task even with only 1 frame as input, likely comes from that the adapter branch learns a different representation compared to the backbone, and the ensemble of the two branches is slightly complementary.
> > >
> > > **Changing the radar graph**
> > >
> > > To ensure a fairer comparison, we have updated the figure to compare models of the same size. We greatly appreciate your suggestion.
> > >
> > >
> > > We hope this clarifies the reviewer's question on model performance on manipulation tasks.
> > >
> > > [1] Learning transferable visual models from natural language supervision, Radford et al., ICML2021
> > >
> > > [2] Learning Manipulation by Predicting Interaction, Zeng et al., RSS2024
> > >
> > > [3] Language-driven representation learning for robotics, Karamcheti et al., RSS2023
> > >
> > > [4] Real-World Robot Learning with Masked Visual Pre-training, Radosavovic et al., PMLR2023
> > >
> > > [5] Feichtenhofer C, Fan H, Malik J, et al. Slowfast networks for video recognition[C]//Proceedings of the IEEE/CVF international conference on computer vision. 2019: 6202-6211.
> > >
> > > [6] Kim M J, Pertsch K, Karamcheti S, et al. OpenVLA: An Open-Source Vision-Language-Action Model[J]. arXiv preprint arXiv:2406.09246, 2024.
> > >
> > > [7] R3M: A Universal Visual Representation for Robot Manipulation, Nair et al, CoRL2022

---

> ### Comment · Reviewer_PuQN · 2024-11-28
>
> Thanks to the authors for their clarifications. Regarding the experiments on robotic manipulation tasks, I maintain my concerns after careful consideration. The results suggest that the combination of EgoVideo(b) + Ego4D, which may approximate EgoVLP/LaViLa/AVION, competes with robot learning methods like R3M, MVP, and Voltron, achieving surprisingly strong performance. It is weird to me that no prior work has discovered this phenomenon, especially considering that both EgoVLP and R3M were published over two years ago. **Given my limited expertise in robotics, I believe the Area Chair could further evaluate this aspect.**
>
> Currently, I am inclined to retain my borderline accept (BA) score.

---

> > ### Author Response · Authors · 2024-11-28
> >
> > We greatly appreciate the reviewer's comment and the continued engagement with our work.
> >
> > We believe we can address the reviewer's concern about the experiments of the robot manipulation task.
> > We hope to provide some information to help address the reviewer's comment on **"It is weird to me that no prior work has discovered this phenomenon, especially considering that EgoVLP was published over two years ago"**. We also provide some explanations of our model's performance.
> >
> > In fact, R3M (released in 2022, also two years ago), first explores using Ego4D data in robot manipulation tasks, and already sees great improvement over CLIP. After that, MVP and Voltron added Masked image modeling or language generation into the pretraining objective, and got some performance increase. Note that, these methods use **image backbones** instead of **video backbones**. We believe the main reason is that current simulators and real robots mainly use image-based observations.
> >
> > Based on our analysis, there are two main reasons that our EgoVideo(b) + Ego4D can already compete with robot learning methods R3M, MVP, and Voltron.
> > - Our EgoVideo is the first **video backbone** applied to the robot manipulation task. Our pretraining injects video information (like temporal motion) into the backbone, enhancing the learned representation.
> > - The Ego4D data we use here come from EgoVLP, which higher in quality because of the filtering by several criteria.
> >
> > Because of this result, we strongly believe that the potential of video-based backbones is a promising new direction for robot manipulation tasks.
> >
> > While we are excited by these findings, we would like to emphasize that the robotic manipulation evaluation is a by-product of our research. The primary focus of our work is egocentric video understanding, as reflected in our title. The robotics experiment serves to demonstrate the broader applicability of our approach and to encourage the exploration of video-based backbones in embodied AI domains.
> >
> > We sincerely hope that this additional context provides clarity and helps address the reviewer’s concerns. We respectfully encourage the reviewer to consider increasing the score, as our work introduces a novel direction for both egocentric video understanding and robotic manipulation through the use of video backbones. Thank you again for your valuable feedback and for recognizing the potential impact of our contributions.

---

### Official Review · Reviewer_xrMB · 2024-11-01

**Soundness:** 3
**Presentation:** 3
**Contribution:** 3
**Rating:** 6
**Confidence:** 4

**Summary:**

The paper presents a new method for egocentric video representation learning that integrates fine-grained hand-object dynamics. It introduces the HOD data pipeline for generating detailed captions and the EgoVideo model, which uses a lightweight motion adapter for efficient learning. The approach achieves state-of-the-art performance across various tasks.

**Strengths:**

1. The HOD pipeline effectively enriches video annotations with hand-object dynamics, capturing interactions more realistically than prior approaches.
2. By using a lightweight motion adapter and co-training strategy, EgoVideo is effective in capturing detailed movements.

**Weaknesses:**

1. Adding a figure to introduce the “Motion Adapter” would make it more intuitive.
2. The proposed method lacks comparison with other methods in terms of parameter and computational complexity.

**Questions:**

1. What is the inference process of the proposed method? If it still relies on both the hand-object detector and LLM as in training, will it significantly increase inference time?
2. The HOD framework uses the original caption as input; what should be done if the dataset lacks corresponding annotation information?

---

> ### Author Response · Authors · 2024-11-21
> **Response to Reviewer xrMB**
>
> **Motion Adapter.** We thank the reviewer's suggestion and have added a figure to introduce our motion adapter. The added figure is Figure 4 of the revised manuscript.
>
> **Parameter and computational complexity.** We compare our EgoVideo with other methods[1-3] in parameter and computational complexity in the following tables. EgoVideo achieves SOTA results at a reasonably low parameter scale.
>
> | Method     | Backbone  | Params | EK100-mAP |
> | ---------- | ------ | ------ | ------------ |
> | LaViLa-B      |TSF-B  | 121M    | 30.9            |
> | AVION-B      | ViT-B | 86M    | 32.9           |
> | EgoVideo-B | ViT-B| 112M  | 36.5       |
> | LaViLa-L      | TSF-L | 438M   | 36.1            |
> | AVION-L     | ViT-B| 307M  | 37.6         |
> | EgoVideo-L | ViT-L | 375M   | 41.8        |
> | EgoVideo-G | ViT-G | 1050M   | 47.1         |
>
> This table shows that our EgoVideo model maintains a relatively small parameter count, and even with the addition of the motion adapter, the total remains lower than that of LaViLa, highlighting the efficiency of our approach.
>
> | Method     | Views  | GFLOPs | Extra GFLOPs |
> | ---------- | ------ | ------ | ------------ |
> | ViT-B      | 4×1×3  | 201    | -            |
> | ViT-B      | 16×1×3 | 804    | -            |
> | LaViLa-B      | 16×1×3 | 1432    | -            |
> | EgoVideo-B | 16×1×3 | 1092   | 288          |
> | ViT-L      | 4×1×3  | 1047   | -            |
> | ViT-L      | 16×1×3 | 4146   | -            |
> | LaViLa-L     | 16×1×3 | 4956   | -            |
> | EgoVideo-L | 16×1×3 | 5308   | 1162         |
>
> This table shows the inference speed of our model and other methods. Thanks to our MotionAdapter, the increase in inference time for our model compared to ViT at 16 frames, is only similar to ViT’s inference time at 4 frames.
>
> **Inference process.** I believe this is due to a misunderstanding of our approach. We only utilize a hand-object detector and a large language model (LLM)  in our data generation pipeline, and our inference process follows the same setting as a standard vision-language pre-training (VLP) model, so it has no impact on the inference process.
>
> **Data without annotation.**  We believe it can be partially solved by using the latest vision language models to provide captions for videos. We list several examples such as GPT4o[5](closed source), or open-source models like llava-next-video[4] and videochat[5]. After video captioning, we can use our HOD data pipeline.
>
> [1] Zhao Y, Misra I, Krähenbühl P, et al. Learning video representations from large language models[C]//Proceedings of the IEEE/CVF Conference on Computer Vision and Pattern Recognition. 2023: 6586-6597.
>
> [2] Dou Z Y, Yang X, Nagarajan T, et al. Unlocking exocentric video-language data for egocentric video representation learning[J]. arXiv preprint arXiv:2408.03567, 2024.
>
> [3] Zhao Y, Krähenbühl P. Training a large video model on a single machine in a day[J]. arXiv preprint arXiv:2309.16669, 2023.
>
> [4] Li F, Zhang R, Zhang H, et al. Llava-next-interleave: Tackling multi-image, video, and 3d in large multimodal models[J]. arXiv preprint arXiv:2407.07895, 2024.
>
> [5] Li K, Wang Y, He Y, et al. Mvbench: A comprehensive multi-modal video understanding benchmark[C]//Proceedings of the IEEE/CVF Conference on Computer Vision and Pattern Recognition. 2024: 22195-22206.

---

> > ### Author Response · Authors · 2024-11-29
> >
> > Dear reviewer,
> >
> > As we near the conclusion of the discussion phase, we would like to inquire if our response has effectively addressed your inquiries. In the rebuttal, we have provided explanations point-by-point for all your questions and concerns. Should our explanations have met your expectations, we kindly request your consideration in revising the score accordingly.
> >
> > Should you have any additional comments or concerns, we are more than willing to address them promptly during the response period. We deeply appreciate your constructive and insightful feedback, which has greatly contributed to the refinement of our work.

---

### Official Review · Reviewer_Fnhi · 2024-11-03

**Soundness:** 3
**Presentation:** 3
**Contribution:** 2
**Rating:** 6
**Confidence:** 3

**Summary:**

The paper introduces a new method for learning from egocentric videos by focusing on hand-object interactions, which are vital for understanding these videos. The authors propose HOD (Hand-Object Dynamics), a system that generates detailed video captions using hand-object detectors and large language models. To leverage this enriched data, they present EgoVideo, a model with a lightweight motion adapter and a co-training strategy that effectively captures hand-object dynamics. EgoVideo outperforms other models in various egocentric video tasks and adapts well to applications like robot manipulation. Because my research field is somewhat distant from this article, I can only give a relatively general judgment. For more detailed judgments, please refer to the analysis of other reviewers.

**Strengths:**

1. The data generation pipeline creates high-quality video-language pairs, enhanced with detailed descriptions of hand-object interactions, enabling effective pretraining for video-language tasks.
2. The proposed methods demonstrate promise for real-world applications in augmented reality, robotic manipulation, and personalized assistants by achieving exceptional results in egocentric tasks.

**Weaknesses:**

1. The author should be more careful in writing. One of the most obvious mistakes is that the title of Figure 3 is wrong. Here, the author uses the title in Figure 6.
2. When the author obtains a better language description, they should also present the underlying semantic rules to clarify its rationale. Furthermore, evaluating the description's quality is essential to ensure standardization and scientific rigor.
3. I think the author could do some further qualitative analysis, starting with the good cases and bad cases. These examples could be considered to be put in the main text instead of the appendix to help readers better understand.

**Questions:**

Please see weakness

---

> ### Author Response · Authors · 2024-11-21
> **Response to Reviewer Fnhi**
>
> **Writing error.** Thank you very much for your correction. We have updated the pdf to address this error.
>
> **Language description rationale & quality.** We first explain our data generation process to demonstrate the rationale behind our data pipeline. We use a hand-object detector to obtain the spatial coordinates of the hand and object within the video clip. These coordinates are then normalized to form the hand-object dynamics prompt. Then, we combine this information with the original caption and input it into an LLM to generate new narrations with fine-grained hand-object interaction descriptions. Our rationale is based on two key points:
>
>   1. The LLM can understand the hand and object position coordinates we provide, thereby extracting motion and interaction dynamics.
>   2. The LLM can integrate this information with the high-level original narration to generate semantically richer and more detailed narrations.
>
> To demonstrate the quality of our data, we conduct the following three experiments.
>
> 1. We analyze the distribution of the words of our HOD against the original EgoClip narrations. We select the top-30 most frequent words, and show their normalized frequency in Figure 3 in our submission. From this figure, we observe that words in the EgoClip narrations are more long-tailed, while our HOD captions show better distribution. Also, from the words, we can see a lot of "dynamic" words in our HOD data like "up" and "downwards". This from one angle proves our rationale in data generation.
>
> 2. We compare our HOD data with another data generation technique: LaViLa-Narrator[1]. We evaluate the CIDEr[2] and METEOR metric using the original human narration as the ground truth. From the table below, we can observe that compared with LaViLa-narrator, our CIDEr score is higher, while the METEOR score is slightly lower. This indicates that our narrations are semantically close to the original captions, preserving the original information. The lower METEOR score, we believe, is due to our narrations incorporating more fine-grained dynamic information, which is not present in the original captions.
>
>   | Text  | METEOR | CIDEr |
>   | --------------- | ------ | ----- |
>   | LaViLa-Narrator | 0.45   | 0.34  |
>   | HOD(ours)       | 0.39   | 0.40  |
>
>   3. To further verify the quality of our HOD data, we employ GPT-4o[3] for quality assessment. We randomly select 1000 clips(due to time and cost) and let GPT evaluate the score of the caption data for the video clip in a range from 0 to 10. To ensure GPT does not simply assign high scores based on the length of the captions, we also conduct random gerund replacements on our data for comparison. The results, summarized in the table below (also Table 1 in our submisstion), show that our HOD data have a significantly better GPT-Score.
>
> | Data            | GPT-Score |
> | --------------- | ------ |
> | EgoClip | 5.53   |
> | HOD-random      | 3.70  |
> | HOD      | 7.71   |
>
> We hope these experiments and explanations can address the reviewers concern.
>
> **Further qualitative analysis.** We thank the reviewer for the constructive suggestion. In Figure 8 of our submission, we show several examples for qualitative analysis (indeed in the appendix). Due to the page limitation, we cannot add a comprehensive figure showing many examples. Alternatively, we added in the main text a pointer to the corresponding figure in the appendix. If the reviewer believes a figure in the main text would be more beneficial, we will add the figure in the revised manuscript.
>
> [1]Zhao Y, Misra I, Krähenbühl P, et al. Learning video representations from large language models[C]//Proceedings of the IEEE/CVF Conference on Computer Vision and Pattern Recognition. 2023: 6586-6597.
>
> [2]Maluuba. nlg-eval. https://github.com/Maluuba/nlg-eval. Accessed: 2022-06-01.
>
> [3]Achiam J, Adler S, Agarwal S, et al. Gpt-4 technical report[J]. arXiv preprint arXiv:2303.08774, 2023.

---

> > ### Comment · Reviewer_Fnhi · 2024-11-26
> >
> > Thank you for the responses. While the work shows promise in egocentric video learning, significant gaps remain in novelty, dataset robustness, and evaluation depth. Reliance on existing frameworks and unbalanced comparisons weaken the impact, and the HOD pipeline’s temporal consistency needs further validation. I encourage refining dataset construction, architectural innovation, and broader evaluations to enhance the contribution. I will maintain my score of **6** as the work is promising but requires substantial improvements.

---

> > > ### Author Response · Authors · 2024-11-27
> > >
> > > We sincerely appreciate the reviewer’s prompt and constructive feedback.
> > >
> > > However, we are unclear about the specific concerns regarding "reliance on existing frameworks" and "unbalanced comparisons." If the reviewer provide more details, we can surely revise our work to meet the reviewer's expectation. This additional information would greatly assist us in revising our work to better align with the reviewer's expectations.
> > >
> > > Additionally, we would appreciate further clarification on the comment regarding the "HOD pipeline’s temporal consistency needing further validation." We believe the reviewer's opinion is valuable and wish to adequately address the reviewer's concerns.
> > >
> > > Thank you once again for your valuable feedback.

---

### Official Review · Reviewer_bLce · 2024-11-04

**Soundness:** 2
**Presentation:** 3
**Contribution:** 2
**Rating:** 6
**Confidence:** 4

**Summary:**

This paper aims to enhance the understanding of complex hand-object interactions in egocentric video-text learning. To accomplish this, the authors introduce a pipeline called HOD, which combines a hand-object detector with a large language model (LLM) to generate high-quality, detailed descriptions of hand-object dynamics. Additionally, they propose a model named EgoVideo, which utilizes a lightweight motion adapter to capture fine-grained hand-object motion information. Through a co-training strategy, EgoVideo effectively and efficiently leverages the detailed hand-object dynamics provided by HOD data. Extensive experiments demonstrate that this approach achieves state-of-the-art performance across various egocentric downstream tasks.

**Strengths:**

This paper has good writing and is easy-to-follow, the idea is straightforward and the experiments shows remarkable improvement. Unlike some previous works who simply rely on extracting additional data of hand-object interaction, this paper provide an effective component (motion adapter) to utilize the data.

**Weaknesses:**

(1) While the motion adapter proves useful, it is not a novel approach, as similar techniques have been explored in numerous prior works. This does not lower my current rating, but it is the primary reason I have not assigned a score of 8.
(2) The experiments could be more comprehensive. For instance, it would be insightful to see performance results without any adapter, using only a single branch. Additionally, I am interested in the model's performance on more challenging tasks, such as robotic manipulation. Providing such additional results could positively impact my rating.

**Questions:**

Please refer to my weakness.

---

> ### Author Response · Authors · 2024-11-21
> **Response to Reviewer bLce**
>
> **Motion adapter has been explored in prior works.** As the reviewer pointed out, the idea of Adapters has been used in many previous works[1]. While this is not an entirely new technique, we believe our Motion Adapter distinguishes itself from other works in the following aspects. 1) Our motion adapter is the first adapter for motion sensing in video-based backbones. 2) The architectural design of our motion adapter is specifically optimized to capture fine-grained hand-object dynamics. Thus, we hope the reviewer could re-consider before making up the final decision.
>
> **More Ablation Experiments.** We thank the reviewer for the suggestion. Without the motion adapter, our EgoVideo-B and EgoVideo-L models are equivalent to ViT-B and ViT-L, respectively. Since our motion adapter can effeciently leverage more frames, we show the performance of one-branch (ViT-B) trained using both 4 frames and 16 frames. Due to the time and computation cost, we do not train a 16 frame ViT-L. Results can be found in the Table below. Our two-branch full EgoVideo model can outperform one-branch model even trained using 16 frames as inputs. As can be seen in Table 7 of the submission, our 2-branch EgoVideo is much more efficient, while outperforming the 16-frame one-branch counterpart.
> | Model                | EK-100 MIR(mAP) | EK-100 MIR(nDCG) | EK-100 CLS(Top1 acc) | EK-100 CLS(Top5 acc) | EGTEA(Mean acc) | EGTEA(Top1 acc) | EgoMCQ(Intra) | EgoMCQ(Inter) |
> | -------------------- | --------------- | ---------------- | -------------------- | -------------------- | --------------- | --------------- | ------------- | ------------- |
> | one-branch-B-16f     | 36.2            | 34.3             | 21.8                    | 41.8                   | 43.5               | 49.5               | 63.6            | **95.2**             |
> | one-branch-B-4f      | 34.5            | 33.3             | 20.3                 | 39.8                 | 41.4            | 46.8            | 61.8          | 94.9          |
> | two-branch-B-(4+16)f | **36.5**            | **34.5**             | **22.4**                 | **43.3**                 | **43.6**            | **51.0**            | **64.6**          | 95.0          |
> | one-branch-L-4f      | 39.7            | 35.9             | 22.6                 | 44.3                 | 44.4            | 49.5            | 63.3          | 94.9          |
> | two-branch-L-(4+16)f | **41.8**            | **37.0**             | **24.0**                 | **46.8**                 | **47.1**           | **51.7**           | **65.5**          | **95.9**          |
>
> **Experiments on robot manipulation tasks.** In our submission, we tested our EgoVideo on the Franka Kitchen dataset designed for the robotic manipulation task. Results show that while not specifically designed for robot manipulation, our model can outperform specialized robotic representation learning models like MVP[2] and Voltron[3], and is comparable to the SOTA method MPI[4]. More details can be found in Table 10 and Figure 7 in our submission. We appreciate the reviewer's suggestion, and will certainly explore applying the model to additional robot manipulation benchmarks, such as Metaworld and Calvin.
>
> [1]Pan J, Lin Z, Zhu X, et al. St-adapter: Parameter-efficient image-to-video transfer learning[J]. Advances in Neural Information Processing Systems, 2022, 35: 26462-26477.
>
> [2]Radosavovic I, Xiao T, James S, et al. Real-world robot learning with masked visual pre-training[C]//Conference on Robot Learning. PMLR, 2023: 416-426.
>
> [3]Karamcheti S, Nair S, Chen A S, et al. Language-driven representation learning for robotics[J]. arXiv preprint arXiv:2302.12766, 2023.
>
> [4]Zeng J, Bu Q, Wang B, et al. Learning Manipulation by Predicting Interaction[J]. arXiv preprint arXiv:2406.00439, 2024.

---

> > ### Comment · Reviewer_bLce · 2024-11-24
> >
> > The authors have adequately addressed my concerns, and my final rating is still **weak accept**. I appreciate that this paper includes extensive experiments to support its claims, and the improvements in performance are noteworthy.
> > However, the primary reason I have not given a higher rating is the lack of novelty. Similar works have been published several times over the past year, and without a breakthrough contribution, it is challenging for me to provide a more favorable evaluation.

---

### Meta-Review · Area_Chair_9FZM · 2024-12-20

**Metareview:**

This paper presents an approach for representation learning for hand-object interactions in egocentric videos. The main idea is to use a hand-object detector and an LLM model to generate detailed annotations for hand-object dynamics. The new dataset, called HOD, consisting of selected videos from Ego4D and How-InterLink7M and generated narrations, is then used to pre-train EgoVideo, for learning hand-object dynamics. The learned representations show strong improvements across different downstream tasks: hand-object interaction and robot manipulation tasks.

* Main reasons to accept the paper include: (i) Its strong performance of EgoVideo on hand-object interaction related tasks on egocentric video understanding; (ii) The new dataset, HOD, with generated narrations, which show to be useful for HOI representation learning.

* Reasons to reject the paper: (i) lack of novelty, i.e., no novel component, but the whole solution including EgoVideo and the dataset HOD, is its main novelty; (ii) Not sure if any videos are leaked to HOD from downstream datasets such as EK-100 and EGTEA (Ego4D may be not but not sure about the How-InterLink7M subset).

**Additional Comments On Reviewer Discussion:**

All reviewers rate the paper above the acceptance threshold (6), but they still concern about the novelty of the work. AC reads all reviews, discussions and the paper. AC recommends to accept the paper as the significant improvements and new dataset contribution may out-weight the concern about novelty. Furthermore, the robust of the representation to other domain such as robot manipulation is a plus.

---

### Decision · Program_Chairs · 2025-01-22

Accept (Poster)